# Multivalency transforms SARS-CoV-2 antibodies into ultrapotent neutralizers

Edurne Rujas[1,2,3], Iga Kucharska[1], Yong Zi Tan [1], Samir Benlekbir[1], Hong Cui[1], Tiantian Zhao[4], Gregory A. Wasney[1,5], Patrick Budylowski[6,7], Furkan Guvenc[6,8], Jocelyn C. Newton[1], Taylor Sicard[1,2], Anthony Semesi[1], Krithika Muthuraman[1], Amy Nouanesengsy[1,2], Clare Burn Aschner [1], Katherine Prieto[1], Stephanie A. Bueler[1], Sawsan Youssef[9], Sindy Liao-Chan[9], Jacob Glanville[9], Natasha Christie-Holmes[6], Samira Mubareka[10,11], Scott D. Gray-Owen [8], John L. Rubinstein [1,2,12], Bebhinn Treanor [4,13,14] & Jean-Philippe Julien [1,2,4✉]

SARS-CoV-2, the virus responsible for COVID-19, has caused a global pandemic. Antibodies can be powerful biotherapeutics to fight viral infections. Here, we use the human apoferritin protomer as a modular subunit to drive oligomerization of antibody fragments and transform antibodies targeting SARS-CoV-2 into exceptionally potent neutralizers. Using this platform, half-maximal inhibitory concentration (IC$_{50}$) values as low as $9 \times 10^{-14}$ M are achieved as a result of up to 10,000-fold potency enhancements compared to corresponding IgGs. Combination of three different antibody specificities and the fragment crystallizable (Fc) domain on a single multivalent molecule conferred the ability to overcome viral sequence variability together with outstanding potency and IgG-like bioavailability. The MULTi-specific, multi-Affinity antiBODY (Multabody or MB) platform thus uniquely leverages binding avidity together with multi-specificity to deliver ultrapotent and broad neutralizers against SARS-CoV-2. The modularity of the platform also makes it relevant for rapid evaluation against other infectious diseases of global health importance. Neutralizing antibodies are a promising therapeutic for SARS-CoV-2.

[1] Program in Molecular Medicine, The Hospital for Sick Children Research Institute, Toronto, ON, Canada. [2] Department of Biochemistry, University of Toronto, Toronto, ON, Canada. [3] Biofisika Institute (CSIC, UPV/EHU) and Department of Biochemistry and Molecular Biology, University of the Basque Country (UPV/EHU), Bilbao, Spain. [4] Department of Immunology, University of Toronto, Toronto, ON, Canada. [5] The Structural & Biophysical Core Facility, The Hospital for Sick Children Research Institute, Toronto, ON, Canada. [6] Combined Containment Level 3 Unit, University of Toronto, Toronto, ON, Canada. [7] Institute of Medical Science, University of Toronto, Toronto, ON, Canada. [8] Department of Molecular Genetics, University of Toronto, Toronto, ON, Canada. [9] Distributed Bio, South San Francisco, CA, USA. [10] Department of Laboratory Medicine and Pathobiology, University of Toronto, Toronto, ON, Canada. [11] Sunnybrook Health Sciences Centre, Toronto, ON, Canada. [12] Department of Medical Biophysics, University of Toronto, Toronto, ON, Canada. [13] Department of Cell and Systems Biology, University of Toronto, Toronto, ON, Canada. [14] Department of Biological Sciences, University of Toronto Scarborough, Toronto, ON, Canada. ✉email: jean-philippe.julien@sickkids.ca

The continuous threat to public health from respiratory viruses such as the novel SARS-CoV-2 underscores the urgent need to rapidly develop and deploy prophylactic and therapeutic interventions to combat pandemics. Monoclonal antibodies (mAbs) have been used effectively for the treatment of infectious diseases as exemplified by palivizumab for the prevention of respiratory syncytial virus in high-risk infants[1] or Zmapp, mAb114, and REGN-EB3 for the treatment of Ebola[2]. Consequently, mAbs targeting the Spike (S) protein of SARS-CoV-2 have been a focus for the development of biomedical countermeasures against COVID-19. To date, several antibodies targeting the S protein have been identified[3–19] with bamlanivimab being the first antibody approved in the United States by the Food and Drug Administration (FDA) for the emergency treatment of SARS-CoV-2 in November 2020. Receptor binding domain (RBD)-directed mAbs that interfere with binding to angiotensin converting enzyme 2 (ACE2), the receptor for cell entry[20], are usually associated with the highest neutralization potencies[6,18,19].

mAbs can be isolated by B-cell sorting from infected donors, immunized animals, or by identifying binders in preassembled libraries. Despite these methodologies being robust and reliable for the discovery of virus-specific mAbs, identification of the best antibody clone is usually associated with a time-cost penalty. In addition, RNA viruses have higher mutations rates than DNA viruses and such mutations can significantly alter the potency of neutralizing antibodies. Indeed, several studies have already shown a reduction in neutralization potency from convalescent serum and resistance of certain mAbs[21–23] to the more recent B.1.1.7[24], B.1.351[25], and B.1.1.28[26,27] variants of SARS-CoV-2. Hence, there is an unmet need for the development of a platform that bridges antibody discovery and the rapid identification and deployment of highly potent neutralizers less susceptible to viral sequence variability.

The potency of an antibody is greatly affected by its ability to simultaneously interact multiple times with its epitope[28–30]. This enhanced apparent affinity, known as avidity, has been previously reported to increase the neutralization potency of nanobodies[31,32] and of IgGs over Fabs[8,10,16] against SARS-CoV-2. To leverage the full power of binding avidity, we have developed an antibody-scaffold technology using the human apoferritin protomer as a modular subunit to multimerize antibody fragments and propel mAbs into ultrapotent neutralizers against SARS-CoV-2. Indeed, the resulting Multabody molecules can increase potency by up to four orders of magnitude over corresponding IgGs. In addition, we demonstrate the ability of this technology to combine three different Fab specificities to better overcome point mutations in the Spike. The Multabody offers a versatile IgG-like "plug-and-play" platform to enhance antiviral characteristics of mAbs against SARS-CoV-2, and demonstrates the power of avidity as a mechanism to be leveraged against viral pathogens.

## Results

**Avidity enhances neutralization potency**. We used the self-assembly of the light chain of human apoferritin to multimerize antigen binding moieties targeting the SARS-CoV-2 S glycoprotein. Apoferritin protomers self-assemble into an octahedrally symmetric structure with an ~6 nm hydrodynamic radius (Rh) composed of 24 identical polypeptides[33]. The N terminus of each apoferritin subunit points outwards of the spherical nanocage and is therefore accessible for the genetic fusion of proteins of interest. Upon folding, apoferritin protomers act as building blocks that drive the multimerization of the 24 proteins fused to their N termini (Fig. 1a).

First, we investigated the impact of multivalency on the ability of the single chain variable domain VHH-72 to block viral infection. VHH-72 has been previously described to neutralize SARS-CoV-2 when fused to a Fc domain, but not in its monovalent format[31]. The light chain of human apoferritin displaying 24 copies of VHH-72 assembled into monodisperse, well-formed spherical particles (Fig. 1b, c) and showed an enhanced binding avidity to the S glycoprotein (Fig. 1d) in comparison to bivalent VHH-72-Fc. Strikingly, display of VHH-72 on the light chain of human apoferritin achieved a ~10,000-fold increase in neutralization potency against SARS-CoV-2 pseudovirus (PsV) compared to the conventional Fc fusion (Fig. 1e), demonstrating the power of avidity to transform binding moieties into potent neutralizers.

**Multabodies have IgG-like properties**. The Fc confers IgGs in vivo half-life and effector functions through interaction with neonatal Fc receptor (FcRn) and Fc gamma receptors (FcγR), respectively. To confer these IgG-like properties to our multimeric scaffold, we next sought to incorporate both binding moieties and Fc domains. Because a Fab is a hetero-dimer consisting of a light and a heavy chain, and the Fc is a homodimer, we created single-chain Fab (scFab) and single-chain Fc (scFc) polypeptide constructs. scFab and scFc domains were directly fused to the N terminus of the apoferritin protomer. For in vivo proof-of-principle experiments, we generated a species-matched surrogate molecule that consists of mouse light chain apoferritin fusions to a mouse scFab and a mouse scFc (IgG2a subtype). Binding kinetics showed that the resulting MB molecule binds mouse FcRn in a pH dependent manner—binding at endosomal pH (5.6) and no binding at physiological pH (7.4)—similar to the parental IgG (Fig. 2a). Expectedly, binding to the high-affinity mouse FcγR1 was enhanced through avidity effects in comparison to the parental IgG. Hence, we generated a modified mouse scFc version that includes the FcγR-silencing mutations LALAP to lower Fc binding in a multimeric context (Fig. 2a). Subcutaneous administration of MBs in C57BL/6 or BALB/c mice was well tolerated with no decrease in body weight or visible adverse events. The MB showed favorable IgG-like serum half-life (Fig. 2b), with a prolonged detectable titer in the sera for the lower FcγR-binding MB (LALAP Fc sequence) compared to the WT MB, indicative of a role for the Fc in dictating in vivo bioavailability. Live 2D and 3D-imaging revealed that the fluorescently-labeled MB biodistributed systemically like the corresponding IgG, without accumulation in any specific tissue (Fig. 2c and Fig. S1). In contrast, 15 nm gold nanoparticles (GNP), which have a similar Rh as MBs, rapidly disseminated from the site of injection (Fig. 2c and Fig. S1). Presumably because all sequences derived from the host, the surrogate mouse MB did not induce an anti-drug antibody response in mice (Fig. 2d), thus further highlighting the IgG-like properties of the MB platform.

**Protein engineering to achieve higher valency**. In view of these favorable results for a mouse MB surrogate, we aimed to generate fully-human MBs derived from the previously reported IgG BD23[12] and IgG 4A8[13] that target the SARS-CoV-2 spike RBD and N-terminal domain (NTD), respectively. Addition of scFcs into the MB reduces the number of scFabs that can be multimerized. In order to endow the MB platform with Fc without compromising Fab avidity and hence neutralization potency, we engineered the apoferritin protomer to accommodate more than 24 components per particle. Based on its four-helical bundle fold, the human apoferritin protomer was split into two halves: the two N-terminal α helices (N-Ferritin) and the two C-terminal α helices (C-Ferritin). In this configuration, the scFc fragment of human IgG1 and the scFab of anti-SARS-CoV-2 IgGs were genetically fused at the N terminus of each apoferritin half,

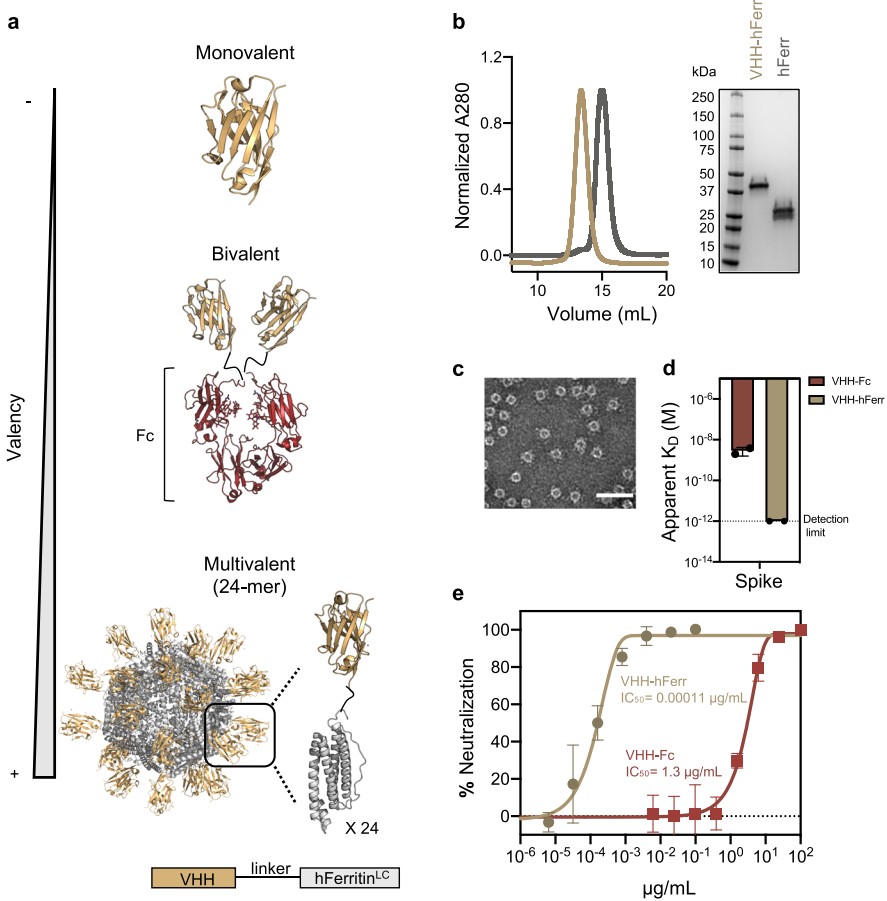

**Fig. 1 Avidity enhances binding and neutralization of VHH against SARS-CoV-2. a** Schematic representation of a monomeric VHH domain and its multimerization using a conventional Fc (dark red) scaffold or human apoferritin (gray). **b** Size exclusion chromatography and SDS-PAGE of apoferritin alone (gray) and VHH-72 apoferritin particles (gold). **c** Negative stain electron microscopy of VHH-72 apoferritin particles. (Scale bar 50 nm, representative of two independent experiments). **d** Comparison of the binding avidity (apparent $K_D$) of VHH-72 to SARS-CoV-2 S protein when displayed in a bivalent (dark red) or 24-mer (gold) format. Bars indicate the mean values of $n = 2$ biologically independent experiments. Apparent $K_D$ lower than $10^{-12}$ M (dash line) is beyond the instrument detection limit. **e** Neutralization potency against SARS-CoV-2 PsV (color coding is as in (**d**)). One representative out of two biologically independent replicates with similar results is shown. Mean values ± SD of two technical replicates is represented in the plot. Median $IC_{50}$ values of the two biologically independent replicates are shown. Source data are provided as a Source Data file.

respectively. Split apoferritin complementation led to hetero-dimerization of the two halves and consequently resulted in a very efficient hetero-dimerization process of the fused proteins. Co-expression of the scFab-C-Ferritin and scFc-N-Ferritin genes together with the scFab-Ferritin gene in excess resulted in a full apoferritin self-assembly that displays high numbers of scFab and low numbers of scFc on the nanocage periphery (Fig. 3a and Materials and Methods). Conveniently, this design allows for the straightforward purification of the MB using Protein A akin to IgG purification.

This split MB design forms 16 nm Rh spherical particles with an uninterrupted ring of density and regularly spaced protruding scFabs and scFc (Fig. 3b, c). Hence, the MB is on the lower size range of natural IgMs[34], but packs more weight on a similar size to achieve high multi-valency. Binding kinetics experiments demonstrated that high binding avidity of the MB for the Spike was preserved upon addition of Fc fragments (Fig. 3d). Binding to human FcγRI and FcRn at endosomal pH confirmed that scFc was properly folded in the split MB design. In addition, the LALAP mutations in the scFc lowered the binding affinity to human FcγRI (Fig. 3e), as previously observed with the surrogate mouse MB (Fig. 2a). SARS-CoV-2 PsV neutralization assays with the split design MBs showed that enhanced binding affinity for the Spike translates into an improved neutralization potency in

comparison to their IgG counterparts, with a ~1600-fold and >2000-fold increase for BD23 and 4A8, respectively (Fig. 3f). Combined, this data supported the further exploration of the MB as an IgG-like platform that confers exquisite binding avidity and PsV neutralization across epitopes on different Spike domains.

**From antibody discovery to ultrapotent neutralizers**. We next assessed the ability of the MB platform to transform mAb binders identified from initial phage display screens into potent neu-tralizers against SARS-CoV-2 (Fig. 4a). Following standard bio-panning protocols against the RBD of SARS-CoV-2, 20 human mAb binders with moderate affinities that range from $10^{-6}$ to $10^{-8}$ M were selected (Supplementary Data 1). These mAbs were produced as full-length IgGs and MBs and their capacity to block viral infection was compared in a neutralization assay against SARS-CoV-2 PsV (Fig. 4b and Fig. S2a). Notably, MB expression yields, homogeneity and thermostability was similar to those of the parental IgG (Fig. S3) and the MB enhanced the potency of 18 out of 20 (90%) IgGs by up to four orders of magnitude. The largest increment was observed for mAb 298 which went from a mean $IC_{50}$ of ~0.3 μg/mL as an IgG to 0.0001 μg/mL as a MB. Strikingly, 11 mAbs were converted from non-neutralizing IgGs to neutralizing MBs in the tested concentration ranges. Seven

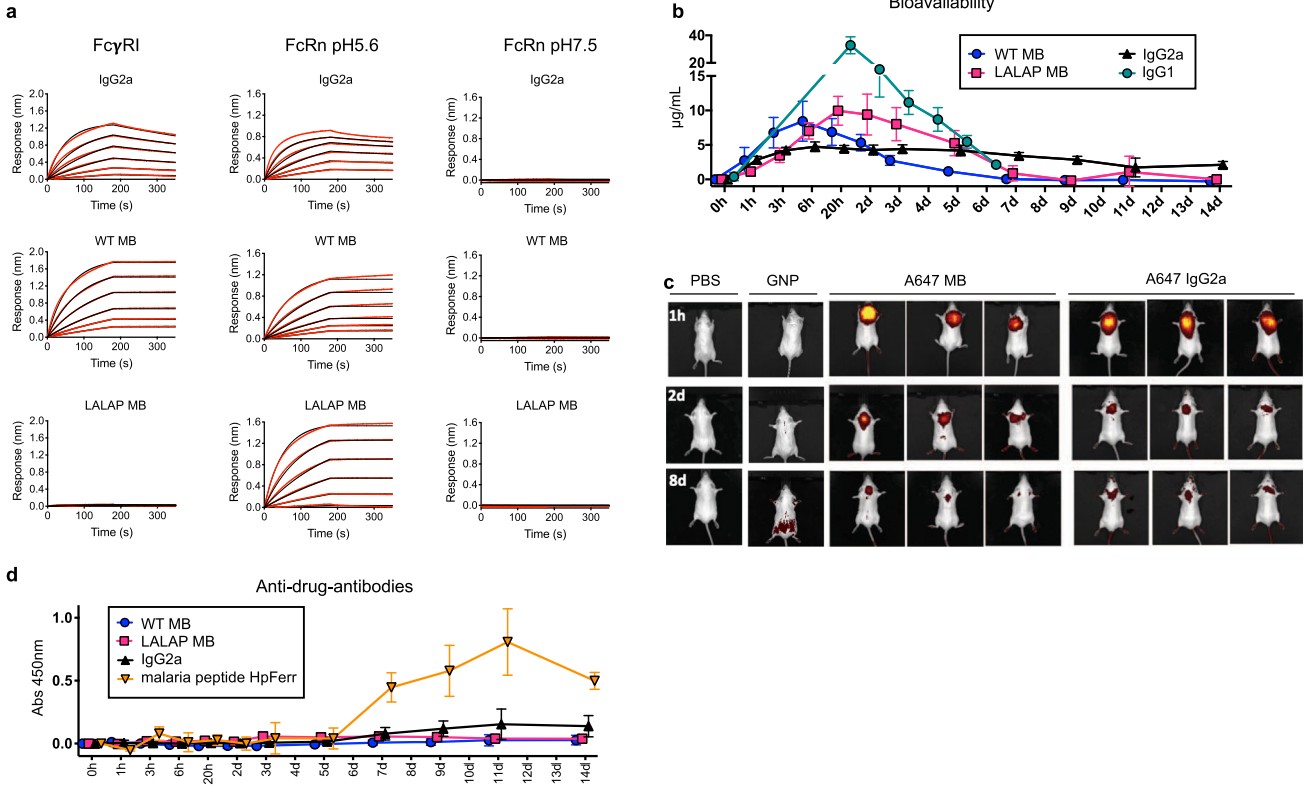

**Fig. 2 Bioavailability, biodistribution, and immunogenicity of a mouse surrogate Multabody. a** Binding kinetics of WT and Fc-modified (LALAP mutation) MB to mouse FcγRI (left) and mouse FcRn at endosomal (middle) and physiological (right) pH in comparison to the parental IgG. Two-fold dilution series from 100 to 3 nM (IgG) and 10 to 0.3 nM (MB) were used. Red lines represent raw data; black lines represent global fits. **b** Five male C57BL/6 mice per group were used to assess the serum concentration of a surrogate mouse MB, a Fc-modified MB (LALAP mutation), and parental mouse IgGs (IgG1 and IgG2a subtypes) after subcutaneous administration of 5 mg/kg. **c** MB and IgG2a samples were labeled with Alexa-647 for visualization of their biodistribution post subcutaneous injection into three male BALB/c mice/group via live noninvasive 2D whole body imaging. 15 nm fluorescently-labeled gold nanoparticles (GNP), which have a similar Rh value as the Multabody are shown as a comparator. **d** Five male C57BL/6 mice per group were used to assess any anti-drug-antibody response induced by the mouse surrogate Multabody in comparison to parental IgG and a species-mismatched malaria PfCSP peptide fused to *Helicobacter pylori* ferritin (HpFerr). Mean values ± SD of $n = 5$ mice is shown in (**b**) and (**d**). Source data of panels **b** and **d** are provided as a Source Data file.

MBs displayed exceptional potency with $IC_{50}$ values between 0.2–2 ng/mL against SARS-CoV-2 PsV using two different target cells (293T-ACE2 and HeLa-ACE2 cells; Fig. 4b and Fig. S2b). PsV neutralization assays using recombinant mAbs REGN10933 and REGN10987 as benchmark showed similar $IC_{50}$ values (0.0044 and 0.030 μg/mL, respectively) to those previously reported[8], and thus confirmed the extraordinary potency of the MBs observed in our assays. The enhanced neutralization potency of the MB was further confirmed with authentic SARS-CoV-2 virus for the mAbs with the highest potency (Fig. 4c and Fig. S2c), as also benchmarked with the two recombinant REGN mAbs. The less sensitive neutralization phenotype we observed against authentic virus in comparison to PsV is also in agreement with previous reports[5,6,9,12].

Retrospectively, all IgGs and MBs were tested for their ability to bind to the Spike glycoprotein and the RBD of SARS-CoV-2 (Fig. S4). Increased avidity resulted in higher apparent binding affinities with no detectable off-rates against the Spike glycoprotein, most likely due to inter-spike crosslinking that translates into high neutralization potency (Fig. 4b–d and Fig. S4). Overall, the data show that the MB platform is compatible with rapid delivery of ultrapotent IgG-like molecules even when starting with mAbs of modest neutralization characteristics.

**Epitope mapping**. Based on their neutralization potency, seven mAbs were selected for further characterization: 298 (IGHV1-46/IGKV4-1), 82 (IGHV1-46/IGKV1-39), 46 (IGHV3-23/IGKV1-39), 324 (IGHV1-69/IGKV1-39), 236 (IGHV1-69/IGKV2-28), 52 (IGHV1-69/IGKV1-39), and 80 (IGHV1-69/IGKV4-1) (Fig. 4b and Supplementary Data 1). Epitope binning experiments showed that these mAbs target two main sites on the RBD, with one of these bins overlapping with the ACE2 binding site (Fig. 5a and Fig. S5). Cryo-EM structures of Fab-SARS-CoV-2 S complexes at a global resolution of ~6–7 Å confirmed that mAbs 324, 298, and 80 bind overlapping epitopes (Fig. 5b, Fig. S6a–c, and Table S1). To gain insight into the binding of mAbs targeting the other bin, we obtained the cryo-EM structure of Fab 46 in complex with the RBD at a global resolution of 4.0 Å (Fig. 5c, Fig. S6d, and Table S1), and the crystal structure of Fabs 298 and 52 as a ternary complex with the RBD at 2.95 Å resolution (Fig. 5d, Fig. S7, and Table S2).

The crystal structure shows that Fab 298 binds almost exclusively to the ACE2 receptor binding motif (RBM) of the RBD (residues 438–506). In fact, out of 16 RBD residues involved in binding Fab 298, 12 are also involved in ACE2-RBD binding (Fig. S7a–c and Table S3). The RBM is stabilized by 11 hydrogen bonds from heavy and light chain residues of Fab 298. In addition, RBM Phe486 is contacted by 11 Fab 298 residues

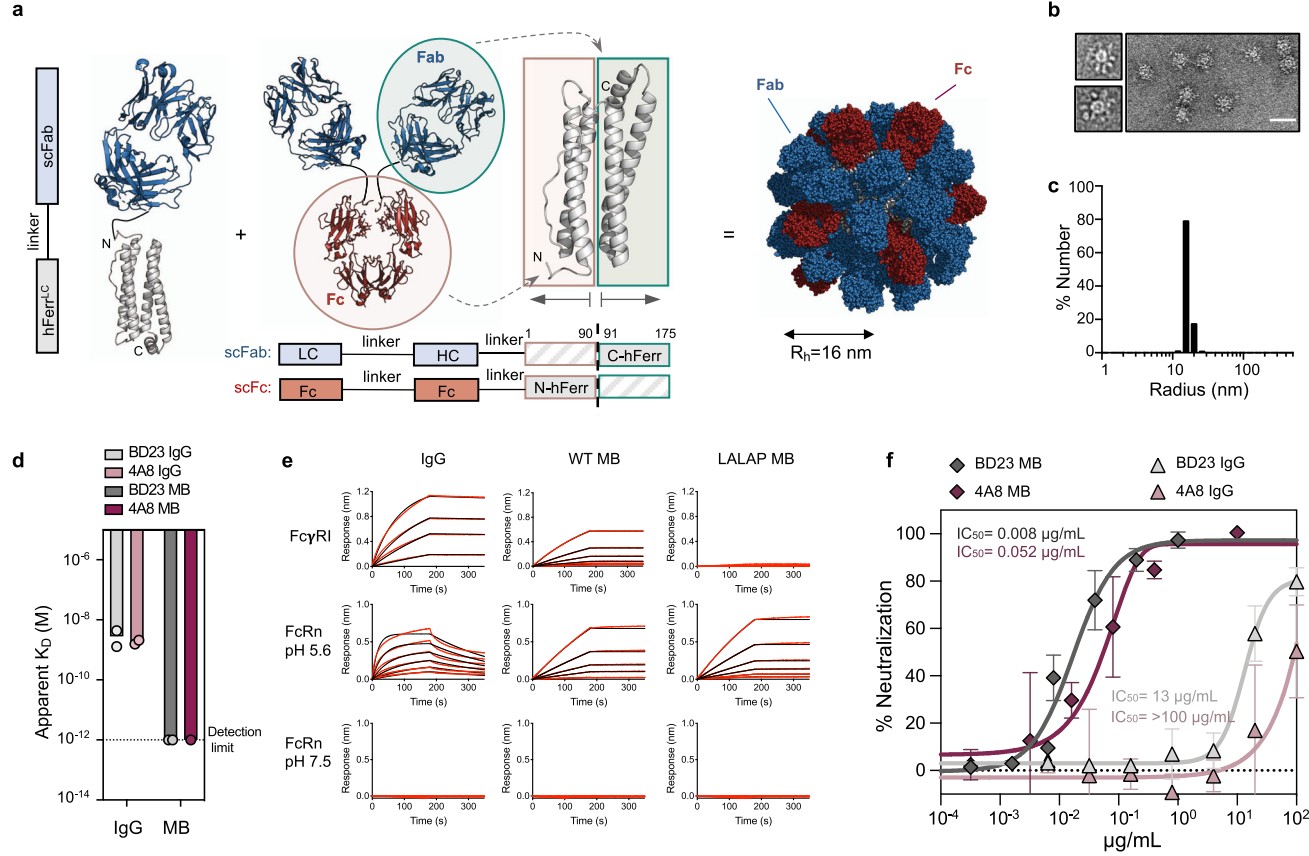

**Fig. 3 Protein engineering to multimerize IgG-like particles against SARS-CoV-2. a** Schematic representation of the human apoferritin split design. **b** Negative stain electron micrograph of the MB. (Scale bar 50 nm, representative of two independent experiments). **c** Hydrodynamic radius (Rh) of the MB. **d** Avidity effect on the binding (apparent $K_D$) of 4A8 (purple) and BD23 (gray) to the SARS-CoV-2 Spike. **e** Sensograms of BD23 IgG and MB with different Fc sequence variants binding to FcγRI (top row), FcRn at endosomal pH (middle row) and FcRn at physiological pH (bottom row). Red lines represent raw data whereas black lines represent global fits. **f** Neutralization of SARS-CoV-2 PsV by 4A8 and BD23 IgGs and MBs. Representative data of three biologically independent samples. The mean values ± SD for two technical replicates is shown in each neutralization plot. Median $IC_{50}$ values of the three biologically independent replicates are indicated. Source data are provided as a Source Data file.

burying ~170 Å² (24% of the total buried surface area on RBD) and hence is central to the antibody–antigen interaction (Fig. S7a and Table S3).

Detailed analysis of the RBD-52 Fab interface reveals that the epitope of mAb 52 is shifted towards the core of the RBD encompassing 20 residues of the RBM and seven residues in the core domain (Fig. 5c, Fig. S7b, and Table S3). In agreement with the competition data, antibody 52 and antibody 46 share a similar binding site, although they approach the RBD with slightly different angles (Fig. 5c, d and Fig. S7d). Inspection of previously reported structures of RBD-antibody complexes reveal that antibodies 46 and 52 target a site of vulnerability on the SARS-CoV-2 spike that has not been described previously (Fig. 5e). The epitope targeted by these antibodies is partially occluded by the NTD in the S "closed" conformation, suggesting that the mechanism of action for this class of antibodies could involve Spike destabilization. Together, these data demonstrate that the enhanced neutralization potency observed for the MB platform through avidity is associated with mAbs that can target distinct epitope bins on the RBD.

**Multabodies overcome Spike sequence variability**. To explore whether MBs could potentially resist viral escape via their enhanced binding avidity, we tested the effect of four naturally occurring RBD mutations[35] on the binding and neutralization of the seven human mAbs of highest potency: L452R—located

within the epitope of antibodies 46 and 52 (bin 1), A475V and V483A—located within the ACE2 binding site (bin 2), and the circulating RBD variant N439K[36] (Fig. 6a–c). In addition, the impact of mutating Asn234—an N-linked glycosylation site—to Gln was also assessed because the absence of glycosylation at this site has been previously reported to decrease sensitivity to neutralizing antibodies targeting the RBD[35]. The more infectious PsV variant D614G[37] was also included in the panel. As expected, mutation L452R significantly decreased binding and potency of mAbs 52 and 46, while antibody 298 was sensitive to mutation A475V (Fig. 6b, c). Deletion of the N-linked glycan at position Asn234 increased viral resistance to the majority of the antibodies, especially mAbs 46, 80, and 324, emphasizing the importance of glycans in viral antigenicity (Fig. 6c). Strikingly, the following antibody specificities in the MB format were minimally impacted in their exceptional neutralization potency by any S mutation: 298, 80, 324, and 236 (Fig. 6d). Mutation L452R decreased the sensitivity of the 46-MB and 52-MB but in contrast to their parental IgGs, they remained neutralizing against this PsV variant (Fig. 6d). The more infectious SARS-CoV-2 PsV variant D614G was neutralized with similar potency as the WT PsV for both IgGs and MBs (Fig. 6c and Fig. S8a).

MB cocktails consisting of three monospecific MBs resulted in pan-neutralization across all PsV variants without a significant loss in potency and hence achieved a 100–1000-fold higher potency compared to the corresponding IgG cocktails (Fig. 6e and

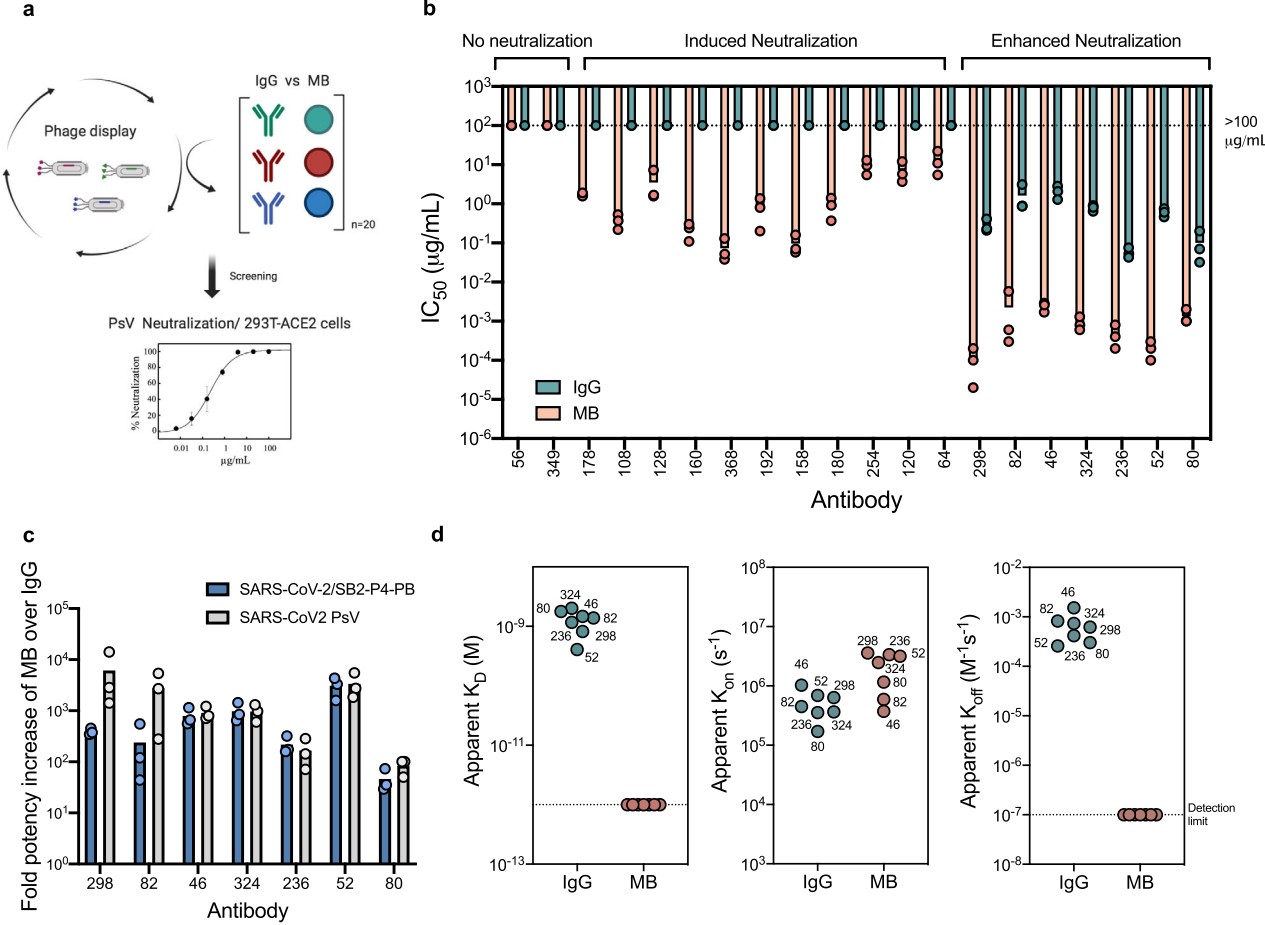

**Fig. 4 The Multabody enhances the potency of human mAbs from phage display. a** Work flow for the identification of potent anti-SARS-CoV-2 neutralizers using the MB technology. Created with Biorender. **b** Comparison of neutralization potency between IgGs (cyan) and MBs (pink) that display the same human Fab sequences derived from phage display. **c** IC$_{50}$ values fold increase upon multimerization. **d** Apparent affinity ($K_D$), association ($k_{on}$), and dissociation ($k_{off}$) rates of the most potent neutralizing MBs (pink) compared to their IgG counterparts (cyan) for binding the SARS-CoV-2 S protein. Three biological replicates and their mean are shown for IC$_{50}$ values in (**b**) and (**c**). Source data of panels **b**–**d** are provided as a Source Data file.

Fig. S8c, d). In order to achieve breadth within a single molecule, we next generated tri-specific MBs by combining multimerization subunits displaying three different Fabs in the same MB assembly (Fig. S8b). Notably, the resulting tri-specific MBs exhibited pan-neutralization while preserving the exceptional neutralization potency of the monospecific versions including against the B.1.351 PsV variant (Fig. 6e, f and Fig. S8c, d). The highest potency was observed for the 298-324-46 combination (Fig. S8c, e), where the tri-specific MB achieved exceptional potency beyond that observed for some of the most potent IgGs reported to date and that we generated recombinantly from available sequences (Fig. 6g). In addition, the MB format was able to increase the potency of these previously reported highly potent IgGs by a further one to two orders of magnitude against PsV and live replicating SARS-CoV-2 virus (Fig. 6h), thus highlighting the plug-and-play nature of the MB and the ability of multivalency to enhance the neutralization capacity of mAbs across a range of potencies.

## Discussion

In this study, we reveal how binding avidity can be leveraged as an effective mechanism to propel antibody neutralization potency and resistance from viral mutations. To this effect, we used protein engineering to develop a plug-and-play antibody-multimerization platform that increases avidity of mAbs

targeting SARS-CoV-2. The seven most potent MBs have IC$_{50}$ values of 0.2 to 2 ng/mL ($9 \times 10^{-14}$ to $9 \times 10^{-13}$ M) against SARS-CoV-2 PsVs and therefore are, to our knowledge, within the most potent antibody-like molecules reported to date against SARS-CoV-2.

The MB platform was designed to include key favorable attributes from a developability perspective. First, the ability to augment antibody potency is independent of antibody sequence, format or epitope targeted. The modularity and flexibility of the platform was exemplified by enhancing the potency of a VHH and multiple Fabs that target non-overlapping regions on two SARS-CoV-2 S sub-domains (RBD and NTD). Using the MB to enhance the potency of VHH domains could provide particular value to this class of molecules since its small size allows highly efficient multimerization. Second, in contrast to other approaches that enhance avidity through tandem fusions of single chain variable fragments[38,39], MBs do not suffer from low stability and in fact self-assemble into highly stable particles with aggregation temperatures similar to those of their parental IgGs. Third, alternative multimerization strategies like streptavidin[40], verotoxin B subunit scaffolds[41], or viral-like nanoparticles[42] face immunogenicity challenges and/or poor bioavailability because of the absence of a Fc fragment and therefore the inability to undergo FcRn–mediated recycling. The light chain of apoferritin is fully human, biologically inactive, has been engineered to

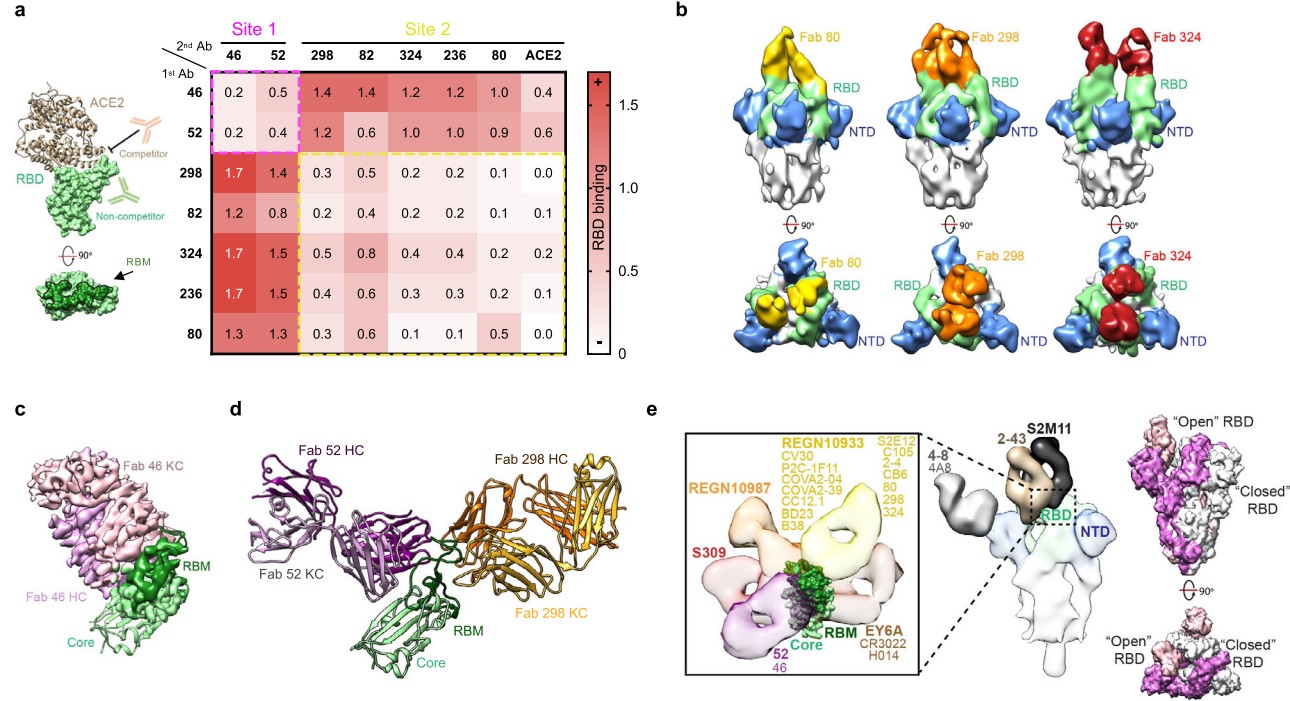

**Fig. 5 Epitope delineation of the most potent mAb specificities. a** Surface and cartoon representation of RBD (light green for the core and dark green for RBM) and ACE2[66] (light brown) binding. Heat map showing binding competition experiments. High signal responses (red) represent low competition while low signal responses (white) correspond to high competition. Epitope bins are highlighted by dashed-line boxes. **b** 15.0 Å filtered cryo-EM reconstruction of the Spike (gray) in complex with Fab 80 (yellow), 298 (orange), and 324 (red). The RBD and NTD are shown in green and blue, respectively. **c** Cryo-EM reconstruction of the Fab 46 (pink) and RBD (green) complex. A RBD[66] secondary structure cartoon is fitted into the partial density observed for the RBD. **d** Crystal structure of the ternary complex formed by Fab 52 (purple), Fab 298 (orange), and RBD (green). **e** Composite image depicting the side and top view of the unliganded (PDB 6XM4) and the antibody-bound SARS-CoV-2 spike with available PDB or EMD entries[3,4,9,10,13,15,17,67–72]. Inset: close up view of antibodies targeting different antigenic sites on the RBD. The mAb with the lowest reported IC$_{50}$ value against SARS-CoV-2 PsV was selected as a representative antibody of the bin (highlighted in bold) and those antibodies with similar binding epitopes are listed in the same color below (color coding of Spike, NTD and RBD as in (**b**)). Individual protomers in the unliganded spike are shown in white, pink, and purple.

include Fc domains, and despite multimerization of >24 Fab/Fc fragments, has a Rh similar to an IgM. As such, a surrogate mouse MB did not elicit antidrug antibodies in mice and similar to its parental IgG was detectable in the sera for over a week. However, in vivo bioavailability of the MB was dependent on its binding affinity to FcγRs, suggesting that Fc avidity will need to be carefully fine-tuned for efficient translation of the MB to the clinic. In addition, further studies will be needed to evaluate how the MB distributes at anatomical sites of interest, such as the lungs in the case of SARS-CoV-2 infection. The plug-and-play nature of the Multabody also lends itself to exploring alternate half-life extending moieties other than the Fc if bioavailability is the only desired trait absent of effector functions e.g., human serum albumin[43], or binding moieties that bind human serum albumin[44,45].

Different increases in neutralization potency were observed for different mAb sequences tested on the MB against SARS-CoV-2. This suggests that the ability of the MB to enhance potency may depend on epitope location on the Spike, or the geometry of how the Fabs engage the antigen to achieve neutralization. The fact that the neutralization of two out of 20 SARS-CoV-2 RBD binders were not rescued by the MB platform suggests limitations based on mAb sequences and binding properties alone. Nevertheless, the capacity of the MB to transform avidity into neutralization potency across a range of epitope specificities on the SARS-CoV-2 Spike highlights the potential for using this

technology broadly. It will be interesting to explore the potency-enhancement capacity of the MB platform against viruses with low surface spike density like HIV-1[46], or against other targets like the tumor necrosis factor receptor superfamily, where bivalency of conventional antibodies limits their efficient activation[47].

Virus escape can arise in response to selective pressure from treatments or during natural selection. A conventional approach to combat escape mutants is the use of antibody cocktails targeting different epitopes. MBs showed a lower susceptibility to S mutations in comparison to their parental IgGs, presumably because the loss in affinity was compensated by enhanced binding avidity. Hence, when used in cocktails, the MB overcame viral sequence variability with exceptional potency. In addition, the split MB design allows combination of multiple antibody specificities within a single multimerized molecule resulting in similar potency and breadth as the MB cocktails. Importantly, the B.1.351 variant of concern that can escape the neutralization of several mAbs[21–23] is neutralized with high potency by a tri-specific Multabody, thus further highlighting the capacity of these molecules to resist viral escape. Multi-specificity within the same particle could offer additional advantages such as intra-S avidity and synergy for the right combination of mAbs, setting the stage for further investigation of different combinations of mAb specificities on the MB. Avidity and multi-specificity could also be leveraged to deliver a single molecule that neutralizes potently across viral genera, such as betacoronaviruses.

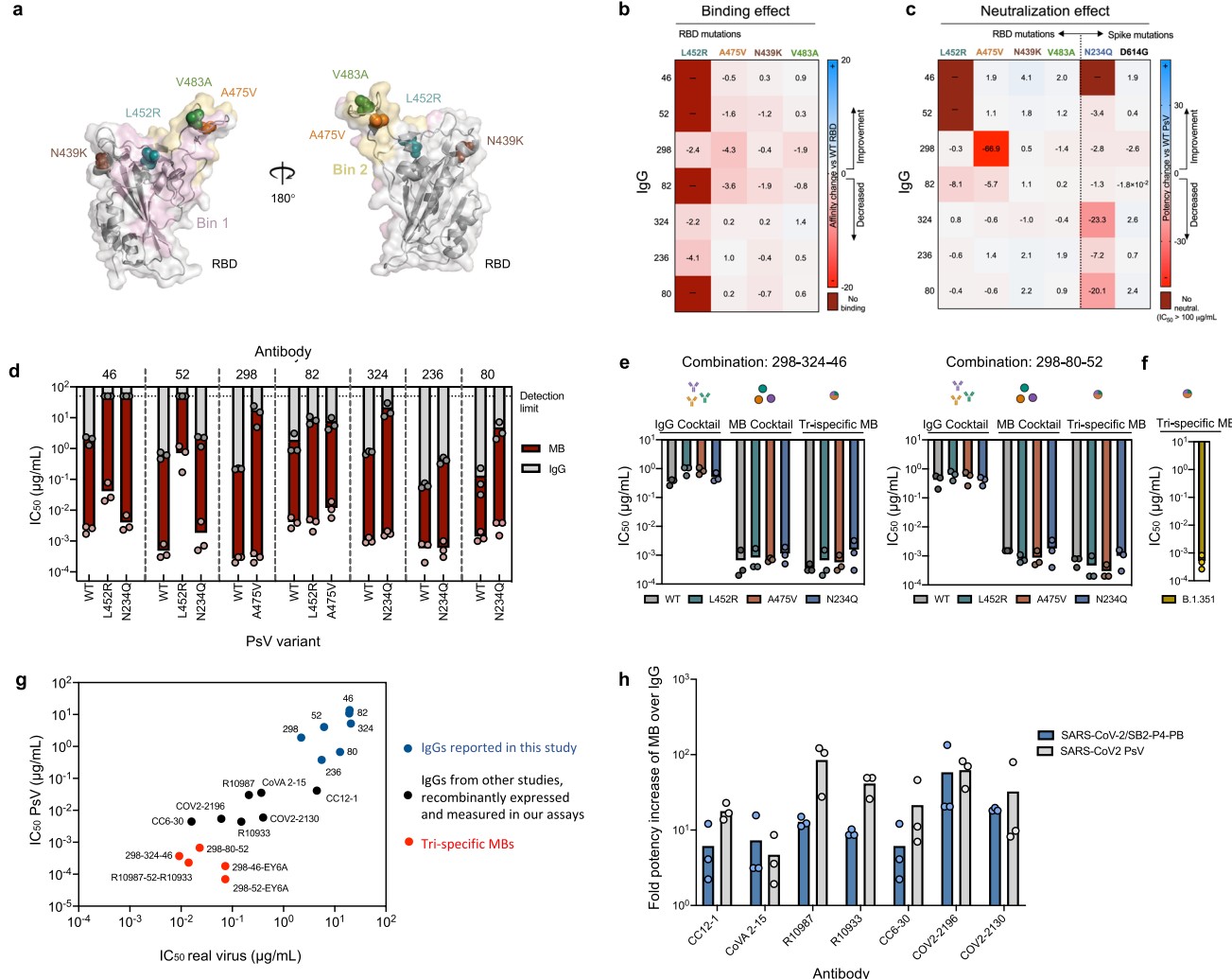

**Fig. 6 Multabodies overcome SARS-CoV-2 sequence diversity. a** Cartoon representation of the RBD showing four naturally occurring mutations as spheres. The epitopes of mAbs 52 (light pink) and 298 (yellow) are shown as representative epitopes of each bin. **b** Affinity and **c** IC$_{50}$ fold-change comparison between WT and mutated RBD and PsV, respectively. **d** Neutralization potency of IgG (gray bars) vs MB (dark red bars) against SARS-CoV-2 PsV variants in comparison to WT PsV. **e** Neutralization potency comparison of two IgG cocktails (three IgGs), monospecific MB cocktails (three MBs) and tri-specific MBs against WT SARS-CoV-2 PsV and variants. mAbs sensitive to one or more PsV variants (**d**) were selected to generate the cocktails and the tri-specific MBs. **f** Neutralization potency of the tri-specific 298-80-52 MB against SARS-CoV-2 B.1.351 PsV variant. **g** IC$_{50}$ values in PsV (y-axis) and replication competent SARS-CoV-2 virus (SB2-P4-PB: x-axis) demonstrating the ability of tri-specific MBs (red) to enhance potency across a wide range of mAb characteristics (blue and black). **h** IC$_{50}$ values fold increase upon multimerization. The mean of three biological replicates is shown in (**b–h**). Source data of panels **d–h** are provided as a Source Data file.

Overall, the MB platform provides a tool to surpass antibody affinity limits and generate broad and potent neutralizing molecules while by-passing extensive antibody discovery or engineering efforts. This platform is an example of how binding avidity can be leveraged to accelerate the timeline to discovery of the most potent biologics against infectious diseases of global health importance.

## Methods

**Protein expression and purification.** Genes encoding VHH-human apoferritin fusion, Fc fusions, Fabs, IgG, and RBD mutants were synthesized and cloned by GeneArt (Life Technologies) into the pcDNA3.4 expression vector. All constructs were expressed transiently in HEK 293F cells (Thermo Fisher Scientific) at a density of $0.8 \times 10^6$ cells/mL with 50 μg of DNA per 200 mL of cells using Fecto-PRO (Polyplus Transfections) in a 1:1 ratio unless specified otherwise. After 6–7 days of incubation at 125 rpm oscillation at 37 °C, 8% CO$_2$, and 70% humidity in a Multitron Pro shaker (Infors HT), cell suspensions were harvested by centrifugation at $5000 \times g$ for 15 min and supernatants were filtered through a 0.22 μm Steritop filter (EMD Millipore). Fabs and IgGs were transiently expressed by co-transfecting 90 μg of the LC and the HC in a 1:2 ratio and purified using

KappaSelect affinity column (GE Healthcare) and HiTrap Protein A HP column (GE Healthcare), respectively with 100 mM glycine pH 2.2 as the elution buffer. Eluted fractions were immediately neutralized with 1 M Tris-HCl, pH 9.0, and further purified using a Superdex 200 Increase size exclusion column (GE Healthcare). Fc fusions of ACE2 and VHH-72 were purified the same way as IgGs. The VHH-72 apoferritin fusion was purified by hydrophobic interaction chromatography using a HiTrap Phenyl HP column and the eluted fraction was loaded onto a Superose 6 10/300 GL size exclusion column (GE Heathcare) in 20 mM sodium phosphate pH 8.0, 150 mM NaCl. Wild type (BEI NR52309) and mutant RBDs, the prefusion S ectodomain (BEI NR52394) and Fc receptors (FcRn and FcγRI) from mouse and human were purified using a HisTrap Ni-NTA column (GE Healthcare). Ni-NTA purification was followed by Superose 6 in the case of the S trimer and Superdex 200 Increase size exclusion columns (GE Heathcare) in the case of the RBD and Fc receptors, in all cases in 20 mM phosphate pH 8.0, 150 mM NaCl buffer.

**Design, expression, and purification of Multabodies.** All molecules referred herein as Multabodies contain scFab and scFc fragments. The scFabs and scFc polypeptide constructs were generated using a 70 amino acid flexible linker [(GGGGS)$_{x14}$] to generate heterodimers and homodimer fragments, respectively. Specifically, the C terminus of the Fab light chain is fused, through the linker, to the

N terminus of the Fab heavy chain. In the case of the scFc, the two single Fc chains that form the functional homodimer Fc were fused in tandem. The individual domains are fused to apoferritin monomers with a 25 amino acid linker: $(GGGGS)_{x5}$. Genes encoding scFab and scFc fragments linked to half apoferritin were generated by deletion of residues 1 to 90 (C-Ferritin) and 91 to 175 (N-Ferritin) of the light chain of human apoferritin. Transient transfection of the Multabodies in HEK 293F cells were obtained by mixing 66 μg of the plasmids scFab-human apoferritin: scFc-human N-Ferritin: scFab-C-Ferritin in a 2:1:1 ratio. Addition of scFab-human apoferritin allowed efficient Multabody assembly and increased the number of Fab's compared to Fc's in the final molecule, thus favoring Fab avidity over Fc avidity. In the case of multi-specific Multabodies, a 4:2:1:1 ratio of scFab1-human apoferritin: scFc-human N-Ferritin: scFab2-C-Ferritin: scFab3-C-Ferritin was used. The DNA mixture was filtered and incubated at room temperature (RT) with 66 μl of FectoPRO before adding to the cell culture. Split Multabodies were purified by affinity chromatography using a HiTrap Protein A HP column (GE Healthcare) with 20 mM Tris pH 8.0, 3 M $MgCl_2$ and 10% glycerol elution buffer. Fractions containing the protein were concentrated and further purified by gel filtration on a Superose 6 10/300 GL column (GE Healthcare).

**Negative-stain electron microscopy**. Three microliters of Multabody at a concentration approximately of 0.02 mg/mL was placed on the surface of a carbon-coated copper grid that had previously been glow-discharged in air for 15 s, allowed to adsorb for 30 s, and stained with 3 μL of 2% uranyl formate. Excess stain was removed immediately from the grid using Whatman No. 1 filter paper and an additional 3 μL of 2% uranyl formate was added for 20 s. Grids were imaged with a FEI Tecnai T20 electron microscope operating at 200 kV and equipped with an Orius charge-coupled device (CCD) camera (Gatan Inc).

**Biolayer interferometry**. Direct binding kinetics measurements were conducted using an Octet RED96 BLI system (Sartorius ForteBio) in PBS pH 7.4, 0.01% BSA, and 0.002% Tween at 25 °C. His-tagged RBD, SARS-CoV-2 Spike was loaded onto Ni-NTA (NTA) biosensors (Sartorius ForteBio) to reach a BLI signal response of 0.8 nm. Association rates were measured by transferring the loaded biosensors to wells containing a two-fold dilution series from 250 to 8 nM (Fabs), 125 to 4 nM (IgG), and 16 to 0.5 nM (MB). Dissociation rates were measured by dipping the biosensors into buffer-containing wells. The duration of each step was 180 s. Fc characterization in the split Multabody design was assessed by measuring binding to hFcγRI and hFcRn loaded onto Ni-NTA (NTA) biosensors following the experimental conditions and concentration ranges indicated above. To probe the theoretical capacity of the Multabodies to undergo endosomal recycling, binding to the hFcRn β2-microglobulin complex was measured at physiological (7.4) and endosomal (5.6) pH. Similarly, Fc characterization of the mouse surrogate MB was assessed by measuring binding to mFcγRI and mFcRn, pre-immobilized onto Ni-NTA (NTA) biosensors. Two-fold dilution series from 100 to 3 nM (IgG) and 10 to 0.3 nM (MB) were used. Analysis of the sensorgrams was performed using the Octet software, with a 1:1 fit model. Competition assays were performed in a two-step binding process. Ni-NTA biosensors preloaded with His-tagged RBD were first dipped into wells containing the primary antibody at 50 μg/mL for 180 s. After a 30 s baseline period, the sensors were dipped into wells containing the second antibody at 50 μg/ml for an additional 300 s. All incubation steps were performed in PBS pH 7.4, 0.01% BSA, and 0.002% Tween at 25 °C. ACE2-Fc was used to map mAb binding to the receptor binding site.

**Dynamic light scattering**. The Rh of the Multabody was determined by dynamic light scattering (DLS) using a DynaPro Plate Reader III (Wyatt Technology). About 20 μL of the Multabody at a concentration of 1 mg/mL was added to a 384-well black, clear bottom plate (Corning) and measured at a fixed temperature of 25 °C with a duration of 5 s per read. Particle size determination and polydispersity were obtained from the accumulation of five reads using the Dynamics software (Wyatt Technology).

**Aggregation temperature**. Aggregation temperature ($T_{agg}$) of the Multabodies and parental IgGs were determined using a UNit instrument (Unchained Labs). Samples were concentrated to 1.0 mg/mL and subjected to a thermal ramp from 25 to 95 °C with 1 °C increments. $T_{agg}$ was determined as the temperature at which 50% increase in the static light scattering at a 266 nm wavelength relative to baseline was observed (i.e., the maximum value of the differential curve). The average and the standard error of two independent measurements were calculated using the UNit analysis software.

**Pharmacokinetics and immunogenicity**. A surrogate Multabody composed of the scFab and scFc fragments of mouse HD37 (anti-hCD19) IgG2a fused to the N-terminus of the light chain of mouse apoferritin (mFerritin) was used for the study. HD37 scFab-mFerritin: Fc-mFerritin: mFerritin in a 2:1:1 ratio was transfected and purified following the procedure described above. L234A, L235A, and P329G (LALAP) mutations were introduced in the mouse IgG2a Fc-construct to silence effector functions of the Multabody[48]. In vivo studies were performed using 12-week-old male C57BL/6 mice purchased from Charles River (Strain code: 027), housed in individually-vented cages under 12 h light/dark cycle (7 a.m./7 p.m.) at a

temperature of 21–23 °C and a humidity of 40–55%. All procedures were approved by the Local Animal Care Committee at the University of Toronto Scarborough. A single injection of ~5 mg/kg of Multabodies or control samples (HD37 single chain IgG-IgG1 or IgG2a subtypes) and *Helicobacter pylori* ferritin (HpFerritin)-PfCSP malaria peptide in 200 μL of PBS (pH 7.5) were subcutaneously injected. Blood samples were collected at multiple time points and serum samples were assessed for levels of circulating antibodies and anti-drug antibodies by ELISA. Briefly, 96-well Pierce Nickel Coated Plates (Thermo Fisher) were coated with 50 μL at 0.5 μg/ml of the $His_{6x}$-tagged antigen hCD19 to determine circulating HD37-specific concentrations using reagent-specific standard curves for IgGs and Multabodies. HRP-ProteinA (Invitrogen) was used to detect the levels of IgG/MBs bound (dilution 1:10,000). For anti-drug-antibody determination, Nunc MaxiSorp plates (Biolegend) were coated with a 12-mer HD37 scFab-mFerritin or with the HpFerritin-PfCSP malaria peptide. 1:100 sera dilution was incubated for 1 h at RT and further develop using HRP-ProteinA (Invitrogen) as a secondary molecule (dilution 1:10,000). The chemiluminescence signal at 450 nm was quantified using a Synergy Neo2 Multi-Mode Assay Microplate Reader (Biotek Instruments).

**Biodistribution**. Eight-week-old male BALB/c mice were purchased from The Jackson Laboratory and housed in individually-vented caging. Mice were housed 14 h of light/10 h dark with phased in dawn to dusk intensity, maximum at noon at a temperature of 20–21 °C and a humidity of 40–60%. All procedures were approved by the Local Animal Care Committee at the University of Toronto. Multabodies composed of the scFab and scFc fragments of mouse HD37 IgG2a fused to the N-terminus of mouse apoferritin light chain was used for this study. HD37 IgG2a Multabody or control samples (HD37 single chain IgG2a) were fluorescently conjugated with Alexa-647 using Alexa Fluor™ 647 Antibody Labeling kit (Invitrogen) as per the manufacturer's instruction. The 15 nm gold nanoparticles labeled with Alexa Fluor™ 647 were purchased from Creative Diagnostics (GFLV-15). PerkinElmer IVIS Spectrum (PerkinElmer) was used to conduct noninvasive biodistribution experiments. BALB/c mice were injected subcutaneously into the loose skin over the shoulders with ~5 mg/kg of the MB, HD37 IgG2a, or gold nanoparticles in 200 μL of PBS (pH 7.5) and imaged at time 0, 1 h, 6 h, 24 h, 2, 3, 4, 8, and 11 days following injection. Prior to imaging, mice were placed in an anesthesia induction chamber containing a mixture of isoflurane and oxygen for 1 min. Anesthetized mice were then placed in the prone position at the center of a built-in heated docking system within the IVIS imaging system (maintained at 37 °C and supplied with a mixture of isoflurane and oxygen). For whole body 2D imaging, mice were imaged for 1–2 s (excitation 640 nm and emission 680 nm) inside the imaging system. Data were analyzed using the IVIS software (Living Image Software for IVIS). After confirming the fluorescent signal from 2D epi-illumination images, 3D transilluminating fluorescence imaging tomography (FLIT) was performed on regions of interest using a built-in scan field of 3 × 3 or 3 × 4 transillumination positions. A series of 2D fluorescent surface radiance images were taken at various transillumination positions using an excitation of 640 and 680 nm emission. A series of CT scans were also taken at the corresponding positions. A 3D distribution map of the fluorescent signal was reconstructed by combining fluorescent signal and CT scans. Resulting 3D fluorescent images were thresholded based on the 3D images of PBS injected mice taken at the corresponding body positions. Images were mapped to the rainbow LUT in the IVIS software, with the upper end of the color scale set to 50 pmol $M^{-1}$ $cm^{-1}$ for mice injected with gold nanoparticles, and 1 000 pmol $M^{-1}$ $cm^{-1}$ for MB and IgG2a injected mice, to allow for better visualization of biodistribution over the time course. A mouse organ registration feature of the IVIS software was used as a general guideline for assessing the sample body locations from 3D images.

**Panning of phage libraries against the RBD of SARS-CoV-2**. The commercial SuperHuman 2.0 Phage library (Distributed Bio/Charles River Laboratories) was used to identify monoclonal antibody binders to the SARS-CoV-2 RBD. For this purpose, an RBD-Fc-Avi tag construct of the SARS-CoV-2 was expressed in the EXPi-293 mammalian expression system. This protein was subsequently purified by protein G Dynabeads, biotinylated and quality-controlled for biotinylation and binding to ACE2 recombinant protein (Sino Biologics Inc). The SuperHuman 2.0 Phage library ($5 \times 10^{12}$) was heated for 10 min at 72 °C and de-selected against Protein G Dynabeads™ (Invitrogen), M-280 Streptavidin Dynabeads™ (Invitrogen), Histone from Calf Thymus (Sigma), Human IgG (Sigma) and ssDNA-Biotin NNK from Integrated DNA Technologies and DNA-Biotin NNK from Integrated DNA Technologies. Next, the library was panned against the RBD-captured by M-280 Streptavidin Dynabeads™ using an automated protocol on Kingfisher FLEX (Thermofisher). Selected phages were acid eluted from the beads and neutralized using Tris-HCl pH 7.9 (Teknova). ER2738 cells were infected with the neutralized phage pools at $OD_{600} = 0.5$ at a 1:10 ratio and after 40 min incubation at 37 °C and 100 rpm, the phage pools were centrifuged and incubated on agar with antibiotic selection overnight at 30 °C. The rescued phages were precipitated by PEG and subjected to three additional rounds of soluble-phase automated panning. PBST/1% BSA buffer and/or PBS/1% BSA was used in the de-selection, washes and selection rounds.

**Screening of anti-SARS-CoV-2 scFvs in bacterial PPE with SARS-CoV-2 RBD.** Anti-SARS-CoV-2 RBD scFvs selected from phage display were expressed and screened using high-throughput surface plasmon resonance (SPR) on Carterra LSA Array SPR instrument (Carterra) equipped with HC200M sensor chip (Carterra) at 25 °C. A V5 epitope tag was added to the scFv to enable capture via immobilized anti-V5 antibody (Abcam, Cambridge, MA) that was pre-immobilized on the chip surface by standard amine-coupling. Briefly: the chip surface was first activated by 10 min injection of a 1:1:1 (v/v/v) mixture of 0.4 M 1-ethyl-3-(3-dimethylamino-propyl) carbodiimide hydrochloride (EDC), 0.1 M N-hydroxysulfosuccinimide (sNHS) and 0.1 M 2-(N-morpholino) ethanesulfonic acid (MES) pH 5.5. Then, 50 μg/ml of anti-V5 tag antibody prepared in 10 mM sodium acetate pH 4.3 was coupled for 14 min and the excess reactive esters were blocked with 1 M ethanolamine HCl pH 8.5 during a 10 min injection. For screening, a 384-ligand array comprising of crude bacterial periplasmic extracts (PPE) containing the scFvs (one spot per scFv) was prepared. Each extract was prepared at a twofold dilution in running buffer (10 mM HEPES pH 7.4, 150 mM NaCl, 3 mM EDTA, and 0.01% (v/v) Tween-20 (HBSTE)) and printed on the anti-V5 surface for 15 min. SARS-CoV-2 RBD Avi Tev His tagged was then prepared at 0, 3.7, 11.1, 33.3, 100, 37, and 300 nM in 10 mM HEPES pH 7.4, 150 mM NaCl, and 0.01% (v/v) Tween-20 (HBST) supplemented with 0.5 mg/ml BSA and injected as analyte for 5 min with a 15 min dissociation time. Samples were injected in ascending concentration without any regeneration step. Binding data from the local reference spots was used to subtracted signal from the active spots and the nearest buffer blank analyte responses were subtracted to double-reference the data. The double-referenced data were fitted to a simple 1:1 Langmuir binding model in Carterra's Kinetic Inspection Tool (version Oct. 2019). Twenty medium-affinity binders from phage display screening were selected for the present study.

**Pseudovirus production and neutralization.** SARS-CoV-2 pseudotyped viruses (PsV) were generated using an HIV-based lentiviral system[49] with few modifications. Briefly, 293T cells were co-transfected with a lentiviral backbone encoding the luciferase reporter gene (BEI NR52516), a plasmid expressing the Spike (BEI NR52310) and plasmids encoding the HIV structural and regulatory proteins Tat (BEI NR52518), Gag-pol (BEI NR52517), and Rev (BEI NR52519) using BioT transfection reagent (Bioland Scientific) and following the manufacturer's instructions. 24 h post transfection at 37 °C, 5 mM sodium butyrate was added to the media and the cells were incubated for an additional 24–30 h at 30 °C. SARS-CoV-2 Spike mutant D614G was kindly provided by D.R. Burton (The Scripps Research Institute), SARS-COV-2 PsV variant B.1.351 was kindly provided by D.D. Ho (Columbia University) and the rest of the PsV mutants were generated using the KOD-Plus mutagenesis kit (Toyobo, Osaka, Japan) using primers described in Table S4. PsV particles were harvested, passed through 0.45 μm pore sterile filters and finally concentrated using a 100 K Amicon (Merck Millipore Amicon-Ultra 2.0 Centrifugal Filter Units).

Neutralization was determined in a single-cycle neutralization assay using 293T-ACE2 cells (BEI NR52511) and HeLa-ACE2 cells (kindly provided by D.R. Burton; The Scripps Research Institute). Cells were seeded the day before the experiment at a density of 10,000 cells/well in a 100 μl volume. In the case of 293T plates, plates where pre-coated with poly-L-lysine (Sigma-Aldrich). The day of the experiment, 50 μl of serially diluted IgGs and MB samples were incubated with 50 μl of PsV for 1 h at 37 °C. After 1 h incubation, the incubated volume was added to the cells and incubated for 48 h. PsV neutralization was monitored by adding 50 μl Britelite plus reagent (PerkinElmer) to 50 μl of the cells and after 2 min incubation, the volume was transferred to a 96-well white plate (Sigma-Aldrich) and the luminescence in relative light units (RLUs) was measured using a Synergy Neo2 Multi-Mode Assay Microplate Reader (Biotek Instruments). Two to three biological replicates with two technical replicates each were performed. IC$_{50}$ fold increase was calculated as:

$$\text{IgG IC}_{50}(\mu g/mL) \Big/ \text{MB IC}_{50}(\mu g/mL)$$

**Authentic virus neutralization.** VeroE6 cells were seeded in a 96 F plate at a concentration of 30,000/well in DMEM supplemented with 100 U Penicillin, 100 U Streptomycin, and 10% FBS. Cells were allowed to adhere to the plate and rest overnight. After 24 h, fivefold serial dilutions of the IgG and MB samples were prepared in DMEM supplemented with 100 U Penicillin and 100 U Streptomycin in a 96 R plate in quadruplicates (25 μL/well). About 25 μL of SARS-CoV-2/SB2-P4-PB[50] Clone 1 was added to each well at 100TCID/well and incubated for 1 h at 37 °C with shaking every 15 min. After co-culturing, the media from the VeroE6 plate was removed, and 50 μL antibody-virus sample was used to inoculate VeroE6 cells in quadruplicates for 1 h at 37 °C, 5% CO$_2$, shaking every 15 min. After 1 h inoculation, the inoculum was removed and 200 μL of fresh DMEM supplemented with 100U Penicillin, 100U Streptomycin, and 2% FBS was added to each well. The plates were further incubated for 5 days. The cytopathic effect (CPE) was monitored and PRISM was used to calculate IC$_{50}$ values. Three biological replicates with four technical replicates each were performed.

**Cross-linking of Spike protein with Fabs 80, 298, and 324.** About 100 μg of Spike trimer was mixed with 2x molar excess of Fab 80, 298, or 324 in 20 mM

HEPES pH 7.0 and 150 mM NaCl. Proteins were crosslinked by addition of 0.075% (v/v) glutaraldehyde (Sigma Aldrich) and incubated at RT for 120 min. Complexes were purified via size exclusion chromatography (Superose6 Increase 10/300 GL, GE Healthcare), concentrated to 0.5 mg/mL and directly used for cryo-EM grid preparation.

**Cross-linking of Fab 46-RBD complex.** About 100 μg of Fab 46 was mixed with 2x molar excess of RBD in 20 mM HEPES pH 7.0 and 150 mM NaCl. The complex was crosslinked by addition of 0.05% (v/v) glutaraldehyde (Sigma Aldrich) and incubated at RT for 45 min. The cross-linked complex was purified via size exclusion chromatography (Superdex 200 Increase 10/300 GL, GE Healthcare), concentrated to 2.0 mg/ml and directly used for cryo-EM grid preparation.

**Cryo-EM data collection and image processing.** Three microliters of sample was deposited on holey gold grids prepared in-house[51], which were glow-discharged in air for 15 s with a PELCO easiGlow (Ted Pella) before use. Sample was blotted for 6 s with a modified FEI Mark III Vitrobot (maintained at 4 °C and 100% humidity) using an offset of −5, and subsequently plunge-frozen in a mixture of liquid ethane and propane. Data were acquired at 300 kV with a Thermo Fisher Scientific Titan Krios G3 electron microscope and prototype Falcon 4 camera operating in electron counting mode at 250 frames/s. Movies were collected for 9.6 s with 29 exposure fractions, a camera exposure rate of ~5 e$^−$/pix/s, and total specimen exposure of ~44 e$^−$/Å$^2$. No objective aperture was used. The pixel size was calibrated at 1.03 Å/pixel from a gold diffraction standard. The microscope was automated with the EPU software package and data collection was monitored with cryoSPARC Live[52].

To overcome preferred orientation encountered with some of the samples, tilted data collection was employed[53]. For the Spike-Fab 80 complex, 820 0˚ tilted movies and 2790 40˚ tilted movies were collected. For the Spike-Fab 298 complex, 4259 0˚ tilted movies and 3513 40˚ tilted movies were collected. For the Spike-Fab 324 complex, 1098 0˚ tilted movies and 3380 40˚ tilted movies were collected. For the RBD-Fab 46 complex, 4722 0˚ tilted movies were collected. For 0˚ tilted movies, cryoSPARC patch motion correction was performed. For 40˚ tilted movies, Relion MotionCorr[54,55] was used. Micrographs were then imported into cryoSPARC and patch CTF estimation was performed. Templates generated from 2D classification during the cryoSPARC Live session were used for template selection of particles. 2D classification was used to remove junk particle images, resulting in a dataset of 80,951 particle images for the Spike-Fab 80 complex, 203,138 particle images for the Spike-Fab 298 complex, 64,365 particle images for the Spike-Fab 324 complex, and 2,143,629 particle images for the RBD-Fab 46 complex. Multiple rounds of multi-class ab initio refinement were used to clean up the particle image stacks, and homogeneous refinement was used to obtain consensus structures. For tilted particles, particle polishing was done within Relion at this stage and reimported back into cryoSPARC. For the Spike-Fab complexes, extensive flexibility was observed. 3D variability analysis was performed[56] and together with heterogeneous refinement used to classify out the different states present. Nonuniform refinement was then performed on the final set of particle images[57]. For the RBD-Fab 46 complex, cryoSPARC ab initio refinement with three classes was used iteratively to clean up the particle image stack. Thereafter, the particle image stack with refined Euler angles was brought into cisTEM for reconstruction[58] to produce a 4.0 Å resolution map. Transfer of data between Relion and cryoSPARC was done with pyem[59].

**Crystallization and structure determination.** A ternary complex of 52 Fab-298 Fab-RBD was obtained by mixing 200 μg of RBD with 2x molar excess of each Fab in 20 mM Tris pH 8.0, 150 mM NaCl, and subsequently purified via size exclusion chromatography (Superdex 200 Increase 10/300 GL, GE Healthcare). Fractions containing the complex were concentrated to 7.3 mg/ml and mixed in a 1:1 ratio with 20% (w/v) 2-propanol, 20% (w/v) PEG 4000, and 0.1 M sodium citrate pH 5.6. Crystals appeared after ~1 day and were cryoprotected in 10% (v/v) ethylene glycol before being flash-frozen in liquid nitrogen.

Data were collected on the 23-ID-D beamline at the Argonne National Laboratory Advanced Photon Source. The dataset was processed using XDS[60] and XPREP. Phases were determined by molecular replacement using Phaser[61] with CNTO88 Fab as a model for 52 Fab (PDB ID: 4DN3), 20358 Fab as a model for 298 Fab (PDB ID: 5CZX), and PDB ID: 6XDG as a search model for the RBD. Refinement of the structure was performed using phenix.refine[62] and iterations of manual building in Coot[63]. PyMOL was utilized for structure analysis and figure rendering[64]. Access to all software was supported through SBGrid[65]. Representative electron density for the two Fab-RBD interfaces is shown in Fig. S7e, f.

**Reporting summary.** Further information on research design is available in the Nature Research Reporting Summary linked to this article.

## Data availability

The electron microscopy maps have been deposited in the Electron Microscopy Data Bank (EMDB) with accession codes EMD-22738, EMD-22739, EMD-22740, and EMD-22741 (Table S1- https://doi.org/10.1101/2020.10.15.341636). The crystal structure of

the 298-52-RBD complex (Table S2- https://doi.org/10.2210/pdb7K9Z/pdb) is available from the Protein Data Bank under accession PDB ID: 7K9Z. The sequences of the monoclonal antibodies used are provided with this paper (Supplementary Data 1). Additional PDB/EMDB entries were used throughout the manuscript to perform a comparative analysis of the different epitope bins targeted by mAbs. The entries used in this analysis are: REGN10933 (PDB ID: 6XDG), CV30 (PDB ID: 6XE1), C105 (PDB ID: 6XCM), COVA2-04 (PDB ID: 7JMO), COVA2-39 (PDB ID: 7JMP), CC12.1 (PDB ID: 6XC2), BD23 (PDB ID: 7BYR), B38 (PDB ID: 7BZ5), P2C-1F11 (PDB ID: 7BWJ), 2-4 (PDB ID: 6XEY), CB6 (PDB ID: 7C01), REGN10987 (PDB ID: 6XDG), S309 (PDB ID: 6WPS, 6WPT), EY6A (PDB ID: 6ZCZ), CR3022 (PDB ID: 6YLA), H014 (PDB ID: 7CAH), 4-8 (EMDB ID: 22159), 4A8 (PDB ID: 7C2L), and 2-43 (EMDB ID: 22275). All data supporting the findings of this study are available within the paper and its supplementary information files. Requests for material can be directed to Jean-Philippe Julien (jean-philippe.julien@sickkids.ca). All materials and reagents will be available upon establishing a material transfer agreement (MTA). Source data are provided with this paper.

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

## Acknowledgements

We thank F. Krammer for providing the WT SARS-CoV-2 Spike plasmid. We thank J.D. Bloom and A.C. Gingras for access to 293T-ACE2 cells and reagents to make the SARS-CoV-2 PsV. We thank D.R. Burton for providing HeLa-ACE2 cells and the SARS-CoV-2 Spike mutant D614G. We are thankful to D.D. Ho for providing the SARS-COV-2 PsV variant B.1.351. We thank C. Pettus, J. Wang, D. Pineda, and K. Patel for their work panning the SuperHuman 2.0 library against the SARS-CoV-2 RBD and characterizing binders. We are grateful to A. Banerjee and K. Mossman for their contributions to the isolation of SARS-CoV-2 (SARS-CoV-2/SB2-P4-PB Clone 1). We thank J.M. Jorgensen and S. Popa for assistance with the Octet RED96 and Unit Instruments. The following reagents were produced under HHSN272201400008C and obtained through BEI Resources, NIAID, NIH: Vector pCAGGS containing the SARS-related Coronavirus 2, Wuhan-Hu-1 Spike glycoprotein RBD, NR-52309, Vector pCAGGS containing the SARS-related Coronavirus 2, Wuhan-Hu-1 Spike glycoprotein gene (soluble and stabilized), NR-52394. This work was supported by Natural Sciences and Engineering Research Council of Canada discovery grant 6280100058 (J.-P.J.), by operating grant PJ4-169662 from the Canadian Institutes of Health Research (CIHR; B.T. and J.-P.J.), by COVID-19 Research Fund C-094-2424972-JULIEN (J.-P.J.) from the Province of Ontario Ministry of Colleges and Universities, by Bill and Melinda Gates Foundation INV-023398 (J.-P.J.) and by the Hospital for Sick Children Foundation. This research was also supported by the European Union's Horizon 2020 research and innovation program under Marie Sklodowska-Curie grant 790012 (E.R.), by a Hospital for Sick Children Restracomp Postdoctoral Fellowship (I.K. and C.B.A.), by a CIHR Postdoctoral Fellowship (Y.Z.T.), by a NSERC postgraduate doctoral scholarship (T.Z.), by a Vanier Canada Graduate Scholarship (T.S.), by a CIHR Canada Graduate Scholarship— Master's Award (A.N.), by the CIFAR Azrieli Global Scholar program (J.-P.J.), by the Ontario Early Researcher Awards program (J.-P.J.), and by the Canada Research Chairs program (J.L.R., B.T., and J.-P.J.). Cryo-EM data were collected at the Toronto High Resolution High Throughput cryo-EM facility, biophysical data at the Structural & Biophysical Core facility, and biodistribution data at the CFI 3D Facility at University of Toronto, all supported by the Canada Foundation for Innovation and Ontario Research Fund. X-ray diffraction experiments were performed at GM/CA@APS, which has been funded in whole or in part with federal funds from the National Cancer Institute (ACB-12002) and the National Institute of General Medical Sciences (AGM-12006). The Eiger 16 M detector at GM/CA-XSD was funded by NIH grant S10 OD012289. This research used resources of the Advanced Photon Source, a US Department of Energy (DOE) Office of Science user facility operated for the DOE Office of Science by Argonne National Laboratory under contract DE-AC02-06CH11357.

## Author contributions

E.R., B.T., and J.-P.J. conceived the research and designed the experiments; E.R., I.K., Y.Z. T., S.B., H.C., T.Z., G.A.W., P.B., F.G., J.C.N., T.S., A.S., K.M., A.N., C.B.A., K.P., S.A.B., S.Y., and S.L.-C. performed experimental work. J.G., N.C.-H., S.M., and S.D.G.-O. provided critical reagents and expertize. E.R., I.K., Y.C.T., B.T., J.L.R., and J.-P.J. analyzed the data. E.R. and J.-P.J. wrote the manuscript with input from all authors.

## Competing interests

The Hospital for Sick Children has applied for patents concerning SARS-CoV-2 antibodies and the Multabody platform technology that are related to this work. B.T. and J.-P.J. are founders of Radiant Biotherapeutics and are members of its Scientific Advisory Board. S.Y., S.L.C., and J.G. are employees of DistributedBio and may hold shares in DistributedBio. The remaining authors declare no competing interests.
