## [Peer Review File · Nature Communications]

Reviewers' Comments:

Reviewer #1:

Remarks to the Author:

The manuscript by Rujas et al entitled "Multivalency transforms SARS-CoV-2 antibodies into ultrapotent neutralizers" presented a very intriguing platform of exploiting antibody multivalent antigen binding mode to increase the antiviral activity of antibodies. Using a single chain variable domain antibody as model, the authors established that fusion of this single chain variable domain antibody to apoferritin protomer resulted in presenting the antibody in a particle with virtually 24 copies per particle and 1000-fold higher neutralization potency against SARS-CoV2 pseudovirus than conventional Fc fusion. Then the authors have engineered a multibody (MB) consisting of scFab and scFc fusion with apoferritin, which displays binding affinity to both FcγRI and FcRn and good in vivo half-lives in mouse. Subsequently, the authors created a more sophisticated MB to incorporate SARS-CoV2 RBD or NTD specific neutralizing antibody with more Fab portion than the Fc portion by fusing the Fab and Fc into C and N terminal domain of apoferritin, respectively. This resulting MBs are >1000-fold more potent than the parental IgG form antibodies. The authors have isolated a panel of SARS-CoV2 RBD-specific antibodies (N=20) from a phage display library derived from COVID19 patients. These antibodies showed moderate affinity for SARS-CoV2 S protein and are weakly neutralizing or non-neutralizing. The MB versions of these antibodies largely gain binding avidity to CoV2 S protein and increased virus neutralization potency, which highlights the general effect of increased avidity on antiviral activity afforded by the MB platform. Additionally, the authors have done nice job on epitope determination of selected RBD-specific antibodies by cryo-EM and crystallography of the complexes of antibody and S protein or RBD. Furthermore, the authors demonstrated that these antibodies in MB format are able to overcome virus neutralization escape mutations better than conventional antibody, and cocktails of MB antibodies are superior to conventional antibody cocktails. Finally, the authors showed that a few tri-specific MBs are the most potent COV2 neutralizing antibodies against the authentic virus, compared to antibodies in conventional format. The experiment design is well thought out. The manuscript is well written. The authors have done an excellent job to exemplify this approach with SARS-CoV2 mAbs. The innovative design of joining multiple antibody functional moieties including both antigen binding and effector function domains in a nanoparticle could be applicable to many immunotherapeutic agents. Some suggestions are listed as below to improve the clarity of the manuscript.

Major point

- 1) It will be very helpful if the author could describe how the scFAB and scFc is designed in the method section. The natural dimerization functions of these molecules seem to be abolished by mutations to coordinate with the ferritin configuration. However, such information is not clear to the readers.
- 2) Fig.3a, Would the scFAB-C-hFerr and scFc-N-hFerr be sufficient to form the particle? It is not clear why the construct hFerr-Fab is needed here? In this case, when hFerr-Fab is included, would it form particle by itself without incorporating the other components encoded by scFAB-C-hFerr and scFc-N-hFerr? How the incorporation ratio is controlled and verified (24 Fab and 8 Fc)?

Minor point

- 1). Fig. 2, please explain why the binding affinity for FcγRI is measured, while there are a few Fcγ receptors.
- 2). Some text correction for antibody name: replace NTD mAb "A48" with "4A8". Fig.3f, A48 IgG to 4A8 IgG.

Reviewer #2:

Remarks to the Author:

The authors have presented an innovative and novel methodology for greatly increasing binding

strength of various antibodies through multimerization on an engineered apoferritin scaffold (which can also incorporate Fc regions). They have applied this to anti-SARS CoV-2 spike antibodies to permit potent in vitro neutralisation of pseudo- and authentic viral infection of ACE2 expressing cells. The protein engineering, characterisation and panning presented represents a significant body of meticulous work. However, some points which remain regarding the therapeutic deployment of these constructs.

Major points for consideration

1. Because of the large size of the multabodies, the in vivo use would be concerning in systemic delivery, circulation and possible entrapment in capillaries. Multabodies are intended to bind a circulating extracellular target (virus) and multimers might occur. If aggregates could form this could cause embolism and other severe side effects. Since there is a concomitant size increase associated with any avidity increase, in this case the increase is quite extreme. The authors point out the need for further biodistribution studies and particle size is one factor in this behaviour. As the authors point out the multobody Rh is approximately that of IgM, however the morphology is quite different so this comparison is not valid for estimation of biodistribution. It would be important to see the experiment in Fig. 2C repeated with IV administration, as this is presumably how such constructs would have to be used clinically.
2. While nanoparticles with size greater than multabodies have been administered in humans without embolism, anti-SARS CoV-2 multabodies are different in that they are intended to bind a circulating extracellular target. There is a concern here with cross-linking of virus and multobody in circulation and potential formation of large aggregates resulting in capillary blockage. This could be more likely to occur in environments with higher concentrations of virus, and compound the problems of clotting observed in COVID-19 patients. Even a simple in vitro experiment to incubate multabodies/antibodies with virus (pseudo or authentic) and check for the formation of aggregates through e.g. DLS, NTA, EM or other suitable characterisation techniques would be beneficial
3. Subcutaneous injection of the large multobody and IgG2a do indeed show similar biodistribution in that they both stay at the injection site. So while it is true in this case that "the fluorescently-labeled MB biodistributed systemically like the corresponding IgG, without accumulation in any specific tissue", a similar result would likely be obtained with subcutaneous injection of fluorescently labelled other macromolecule. Could the comparison be performed of multobody and IgM that have similar Rh values.

Minor points

1. It is acknowledged that the situation in this aspect of the pandemic is constantly changing but in the introduction should update "and although few variants appear to be currently dominant in circulation" to reflect current situation.
2. For the Fig. 3 split apoferritin construct, how was the ratio of split:unsplit ferritin arrived at?

Reviewer #3:

Remarks to the Author:

In the presented paper of Rujas E. et al., the consortium developed an antibody multimerization platform (Multobody) based on the oligomeric protein ferritin to combine avidity effects with multi-specificity to deliver potent and broad neutralizers against SARS-CoV-2. This data-rich work combines various disciplines from protein engineering and design, to structural and cell biology and is a prime example for interdisciplinary research and might be applicable to other diseases or diagnostic applications.

Overall, the experiments were performed properly, the data are well analyzed and interesting. I have a few points/comments/requests which should be addressed prior to publication:

- 1.) Construct design: Can the author elaborate on the strategy of how the apoferritin-fusion constructs were designed? What kind of linker was used (see Fig 1 and 3)? Is this a very flexible or stiff linker? Did you do any optimization steps for the construct design?

- 2.) Did the authors consider fusing an entire IgG to a ferritin monomer? What is the advantage of the current design? To minimize steric hindrance, could it be an advantage to only have e.g. 12 binding moieties of the surface, or is more always better?
- 3.) The authors state that the MB preparations typically have 24 Fab and 8 Fc presented – what is the experimental evidence for this distribution? Did all MB preparations of all the mentioned constructs always contain an Fc-fraction (this is not clear from the MM section)? What are typical yields of these expressions and how homogeneous are they after purification (please show examples)
- 4.) Numerous VHH high-affinity RBD binders (after immunization or from synthetic libraries) with decent neutralization efficiencies have been described. Can these ones be also formatted to the MB platform and would they have an advantage or disadvantage compared to mAb or Fabs (to be discussed in the discussion section)
- 5.) The structural data are interesting but for a “non-structural biologist” not straight forward to understand. A figure on the trimeric nature of the spike protein could help to guide the reader and illustrate the captured conformation. Furthermore, it is not stated how many Fabs are bound to the spike protein and which configuration the RBDs display (up, or down). Color-code in Fig. 5e is unclear, hard to read and understand. All structure work (X-ray and cryo-EM) has been done with Fabs and not full IgGs – this description is mixed up, please unify. Multiple structures of the Spike protein have been determined to higher resolution described here – could Fab binding lead to the preferred orientation of the complex on the EM grid requiring tilted data collection? Is this commonly seen in cryoEM with Fabs?

Minor points:

- Introduce abbreviations e.g. “MB” as Multibody is not mentioned
- Overview table of tested constructs with the measured affinities and neutralization efficiencies could be useful as a supplementary table
- Binding Curves are called sensorgrams – I recommend adding the fitted parameters to all sensorgrams
- What would be the next step of relevance to make use of the study?

Point-by-point response to the Reviewers' comments

Reviewer #1:

The manuscript by Rujas et al entitled “Multivalency transforms SARS-CoV-2 antibodies into ultrapotent neutralizers” presented a very intriguing platform of exploiting antibody multivalent antigen binding mode to increase the antiviral activity of antibodies. Using a single chain variable domain antibody as model, the authors established that fusion of this single chain variable domain antibody to apoferritin protomer resulted in presenting the antibody in a particle with virtually 24 copies per particle and 1000-fold higher neutralization potency against SARS-CoV2 pseudovirus than conventional Fc fusion. Then the authors have engineered a multibody (MB) consisting of scFab and scFc fusion with apoferritin, which displays binding affinity to both FcγRI and FcγRII and good in vivo half-lives in mouse. Subsequently, the authors created a more sophisticated MB to incorporate SARS-CoV2 RBD or NTD specific neutralizing antibody with more Fab portion than the Fc portion by fusing the Fab and Fc into C and N terminal domain of apoferritin, respectively. This resulting MBs are >1000-fold more potent than the parental IgG form antibodies. The authors have isolated a panel of SARS-CoV2 RBD-specific antibodies (N=20) from a phage display library derived from COVID19 patients. These antibodies showed moderate affinity for SARS-CoV2 S protein and are weakly neutralizing or non-neutralizing. The MB versions of these antibodies largely gain binding avidity to CoV2 S protein and increased virus neutralization potency, which highlights the general effect of increased avidity on antiviral activity afforded by the MB platform. Additionally, the authors have done nice job on epitope determination of selected RBD-specific antibodies by cryo-EM and crystallography of the complexes of antibody and S protein or RBD. Furthermore, the authors demonstrated that these antibodies in MB format are able to overcome virus neutralization escape mutations better than conventional antibody, and cocktails of MB antibodies are superior to conventional antibody cocktails. Finally, the authors showed that a few tri-specific MBs are the most potent COV2 neutralizing antibodies against the authentic virus, compared to antibodies in conventional format. The experiment design is well thought out. The manuscript is well written. The authors have done an excellent job to exemplify this approach with SARS-CoV2 mAbs. The innovative design of joining multiple antibody functional moieties including both antigen binding and effector function domains in a nanoparticle could be applicable to many immunotherapeutic agents. Some suggestions are listed as below to improve the clarity of the manuscript.

We thank the Reviewer for the positive assessment of the manuscript and for acknowledging the broad use of the Multibody platform.

1) It will be very helpful if the author could describe how the scFAB and scFc is designed in the method section. The natural dimerization functions of these molecules seem to be abolished by mutations to coordinate with the ferritin configuration. However, such information is not clear to the readers.

As requested by the Reviewer, we have added the following paragraph on line 597 of the Materials and Methods to better describe the way the scFab and scFc fragments were designed: “The scFabs and scFc polypeptide constructs were generated using a 70 amino acid flexible linker [(GGGS)_{x14}] to generate heterodimers and homodimer fragments,

respectively. Specifically, the C terminus of the Fab light chain is fused, through the linker, to the N terminus of the Fab heavy chain. In the case of the scFc, the two single Fc chains that form the functional homodimer Fc were fused in tandem. The individual domains are fused to apoferritin monomers with a 25 amino acid linker: (GGGGS)_{x5}.”

In addition, Fig. 3A has been slightly updated to better illustrate the design.

2) Fig.3a, Would the scFab-C-hFerr and scFc-N-hFerr be sufficient to form the particle? It is not clear why the construct hFerr-Fab is needed here? In this case, when hFerr-Fab is included, would it form particle by itself without incorporating the other components encoded by scFab-C-hFerr and scFc-N-hFerr? How the incorporation ratio is controlled and verified (24 Fab and 8 Fc)?

We appreciate the questions raised by the Reviewer, and agree with the need of additional explanation to clarify the need of the three components that form the Multabody.

First, in the reported design, we noted that the absence of the hFerr-Fab component during transfection of scFab-C-hFerr and scFc-N-hFerr did not allow for the assembly of particles with 42 cargos displayed on the surface of the apoferritin cage. Indeed, in the absence of the hFerr-Fab component, we observed largely aggregation when scFab-C-hFerr and scFc-hFerr were co-transfected. We postulated that steric hindrance may limit the amount of Fab/Fc that can be oligomerized and consequently, added the scFab-hFerr component. The following sentence has been added in line 607 to provide this clarification: “Addition of scFab-human apoferritin allowed efficient Multabody assembly and increased the number of Fab’s compared to Fc’s in the final molecule, thus favoring Fab avidity over Fc avidity”.

hFerr-Fab has the ability to form particles by itself without the need of co-assembly with the other components scFab-C-hFerr and scFc-N-hFerr, as pointed out by the Reviewer. However, purification using protein A chromatography ensures the presence of the Fc fragment, and thus of scFc-N-hFerr (and consequently scFab-C-hFerr) in the purified molecules.

All the molecules reported in the manuscript were produced by the transient co-transfection of plasmids. Therefore, the incorporation ratio of each component is broadly controlled by the DNA ratios transfected, the abundance of resulting translated products, and the dynamics of their assembly. As indicated in the Materials and Methods section, a 2:1:1 ratio of the DNA encoding for scFab-human apoferritin: scFc-human N-Ferritin: scFab-C-Ferritin was used. This ratio, which theoretically results in 24 Fab’s and 8 Fc’s per Multabody particle was chosen to achieve incorporation of the highest amount of cargo in the Multabody, as well as maximize the yield of monodisperse particles in SEC. The geometry of the split MB design together with the DNA ratios in theory translates into a tight control of the amount of each component in the final molecule. However, in agreement with the Reviewer, due to a lack of experimental methods currently available to confirm these exact numbers in a molecule of ~2.4 MDa, the stated number of each subunit in a Multabody has conservatively been removed from Fig. 3A.

3) Fig. 2, please explain why the binding affinity for FcγRI is measured, while there are a few Fcγ receptors.

The Reviewer is correct; only binding to FcγRI is reported in the manuscript. The reason for choosing this receptor over the others is due to its higher affinity to Fc. In addition to showing functional binding of the Fc fragment on a Multabody, the objective of the experiment shown in Fig. 2 was to generate a Multabody with lower Fc binding avidity by introducing the FcγR-silencing mutations LALAP to counter the avidity effect. Hence, showing ablation of FcγR binding using the highest affinity Fc receptor was considered more relevant to support this argument. This point has been added in line 134 for further clarity: “Expectedly, binding to the high-affinity mouse FcγR1 was enhanced through avidity effects in comparison to the parental IgG. Hence, we generated a modified mouse scFc version that includes the FcγR-silencing mutations LALAP to lower Fc binding in a multimeric context (Fig. 2A).”

4) Some text correction for antibody name: replace NTD mAb “A48 “with “4A8”. Fig.3f, A48 IgG to 4A8 IgG.

We thank the Reviewer for this careful reading of the manuscript. A48 has been modified by 4A8 in lines 175, 432, 438 and in Fig. 3F.

Reviewer #2:

The authors have presented an innovative and novel methodology for greatly increasing binding strength of various antibodies through multimerization on an engineered apoferritin scaffold (which can also incorporate Fc regions). They have applied this to anti-SARS CoV-2 spike antibodies to permit potent in vitro neutralisation of pseudo- and authentic viral infection of ACE2 expressing cells. The protein engineering, characterisation and panning presented represents a significant body of meticulous work. However, some points which remain regarding the therapeutic deployment of these constructs.

We thank the Reviewer for the encouraging comments and appreciation of the novelty of our manuscript. Indeed, evaluation of the therapeutic potential of a SARS-CoV-2 Multabody is an important next milestone, and beyond the scope of the current body of work, which focuses on how multivalency can transform SARS-CoV-2 antibodies into ultrapotent neutralizers.

1. Because of the large size of the multabodies, the in vivo use would be concerning in systemic delivery, circulation and possible entrapment in capillaries. Multabodies are intended to bind a circulating extracellular target (virus) and multimers might occur. If aggregates could form this could cause embolism and other severe side effects. Since there is a concomitant size increase associated with any avidity increase, in this case the increase is quite extreme. The authors point out the need for further biodistribution studies and particle size is one factor in this behaviour. As the authors point out the multabody Rh is approximately that of IgM, however the morphology is quite different so this comparison is not valid for estimation of biodistribution. It would be important to see the experiment in Fig. 2C repeated with IV administration, as this is presumably how such constructs would have to be used clinically.

The Reviewer is correct in pointing out that several in vivo parameters will need to be evaluated in attempts to translate a SARS-CoV-2 Multabody into a therapeutic for clinical use. The biodistribution results presented here are indeed absent any in vivo target and are meant to evaluate how a surrogate mouse Multabody may biodistribute generally upon administration and characterize its half-life kinetics. We appreciate the suggestion of the Reviewer regarding putative differences between IV vs SQ administration of the MB. The reason why SQ administration was chosen for this preliminary report is because we believe that a biologic with exquisitely high potency might be able to be delivered in smaller volumes compatible with SQ administration, and therefore valued this characterization. However, we agree that assessing more deeply the in vivo properties of a relevant SARS-CoV-2 MB (also including prophylactic vs therapeutic MoA, dose, repeat dosing, route of administration, pharmacokinetics with and without target, toxicology, biodistribution, different pre-clinical species, etc) – instead of with a mouse surrogate MB as described here – is a critical aspect of future studies looking to evaluate this technology in pre-clinical development.

2. While nanoparticles with size greater than multabodies have been administered in humans without embolism, anti-SARS CoV-2 multabodies are different in that they are intended to bind a circulating extracellular target. There is a concern here with cross-linking of virus and multabody in circulation and potential formation of large aggregates resulting in capillary blockage. This could be more likely to occur in environments with higher concentrations of virus, and compound the problems of clotting observed in COVID-19 patients. Even a simple in vitro experiment to incubate multabodies/antibodies with virus (pseudo or authentic) and check for the formation of aggregates through e.g. DLS, NTA, EM or other suitable characterisation techniques would be beneficial.

How SARS-CoV-2 Multabodies behave in vivo in the presence of target is indeed an area of ongoing interest. It is however hard to speculate whether some of the theoretical concerns raised by the Reviewer will be observed in the lung, or whether neutralization by a SARS-CoV-2 Multabody will be more aligned with the effective IgM neutralization of virus (mediated by 10-12 Fabs, 6 closely-spaced Fc's and similar hydrodynamic radius) as observed in reports of natural infection (e.g. Gasser R., *Cell Reports* (2021); Atyeo C., *Immunity* (2020)). Noteworthy, the presence of high levels of these avid IgM molecules is not associated with patient severity (e.g. Wu J., *Nature Commun* (2021); Li K., *Nature Commun* (2020)).

During the revision period, we have explored some of the in vitro experiments proposed by the Reviewer to identify the putative formation of any proposed aggregate using DLS and EM. However, these experiments weren't straightforward primarily because of reduced reading sensitivity in the presence of FBS in our viral stocks, which couldn't be efficiently removed. In addition, it is unclear at this point whether any of these preliminary in vitro results could be linked to any associated in vivo phenotype given the lack of true negative and positive controls that bridge the in vitro/in vivo gap with which to compare a SARS-CoV-2 Multabody. Hence, we believe the present question will be better answered in a subsequent study where the in vivo efficacy of SARS-CoV-2 Multabodies will be closely evaluated as outlined above.

3. Subcutaneous injection of the large multibody and IgG2a do indeed show similar biodistribution in that they both stay at the injection site. So while it is true in this case that “the fluorescently-labeled MB biodistributed systemically like the corresponding IgG, without accumulation in any specific tissue”, a similar result would likely be obtained with subcutaneous injection of fluorescently labelled other macromolecule. Could the comparison be performed of multibody and IgM that have similar Rh values.

We thank the Reviewer for the suggestion. We have added new 2D (Fig. 2C) and 3D data (Supplementary Fig. S1) comparing the biodistribution of the mouse surrogate Multabody with 15 nm fluorescently-labeled gold nanoparticles, which have a similar Rh value as the Multabody. Consistent with our 2D data and similar to IgGs, the Multabody dissipates from the site of injection over the first 48 hours, and by day 8 are only weakly detected above background in the lower mid-region of the mice. This signal is further diminished by day 11 post-injection, and undetectable in one mouse, further confirming that the Multabody does not accumulate in any specific tissue. In contrast, gold nanoparticles of similar size rapidly disseminate from the site of injection and are nearly undetectable even 1 hour post-injection. This finding further substantiates that the surrogate mouse Multabody biodistributes similarly to its corresponding IgG, and not similarly to an inorganic particle of similar size. This data is now described in line 141, Fig. 2C and in the new Supplementary Fig. S1.

4. It is acknowledged that the situation in this aspect of the pandemic is constantly changing but in the introduction should update “and although few variants appear to be currently dominant in circulation” to reflect current situation.

We thank the Reviewer for this reminder and following that suggestion, the following sentence in the introduction has been added (line 80): “Indeed, several studies have already shown a reduction in neutralization potency from convalescent serum and resistance to certain mAbs^{16–18} to the more recent B.1.1.7¹², B.1.351¹³ and B.1.1.28^{14,15} variants of SARS-CoV-2.”

Due to the concerns raised by these emerging variants, the neutralization potency of a representative tri-specific Multabody was tested against the B.1.351 PsV variant. Confirming the capacity of these molecules to resist viral escape, when tested against the B.1.351 PsV variant, the tri-specific MB preserved its extraordinary potency. The IC₅₀ value of the MB against the variant of concern was 0.00052 µg/mL and hence no effect in potency was observed in comparison to the WT SARS-CoV-2 PsV (IC₅₀ value of 0.00067 µg/mL). These data have been added to the Results (line 285), Fig. 6F and the following sentence has been added to the Discussion in line 349: “Importantly, the B.1.351 variant of concern that can escape the neutralization of several mAbs^{16–18} is neutralized with high potency by a tri-specific Multabody, thus further highlighting the capacity of these molecules to resist viral escape”.

5. For the Fig. 3 split apoferritin construct, how was the ratio of split:unsplit ferritin arrived at?

We would like to point the Reviewer to our detailed answer above to Reviewer 1's similar question (#2) regarding the ratio of Multabody components.

Reviewer #3:

In the presented paper of Rujas E. et al., the consortium developed an antibody multimerization platform (Multabody) based on the oligomeric protein ferritin to combine avidity effects with multi-specificity to deliver potent and broad neutralizers against SARS-CoV-2. This data-rich work combines various disciplines from protein engineering and design, to structural and cell biology and is a prime example for interdisciplinary research and might be applicable to other diseases or diagnostic applications. Overall, the experiments were performed properly, the data are well analyzed and interesting. I have a few points/comments/requests which should be addressed prior to publication:

We thank the Reviewer for these encouraging comments on the manuscript.

1) Construct design: Can the author elaborate on the strategy of how the apoferritin-fusion constructs were designed? What kind of linker was used (see Fig 1 and 3)? Is this a very flexible or stiff linker? Did you do any optimization steps for the construct design?

We refer the Reviewer to the explanation provided on point #1 for Reviewer 1 above for a detailed clarification on the exact design of the apoferritin fusion constructs, with corresponding changes added to the Materials and Methods section (line 605) and Fig. 3A. As for the linkers of the fusion proteins, screening of different lengths revealed that assembly of scFab's and scFc's was efficient with 70 amino acids linkers, as previously reported (Koerber J.T., J Mol Biol (2015). Similarly, we studied the effect of the linker length that connect the cargo (scFab and scFc) to the apoferritin scaffold and found that the high expression yields were obtained with 25 amino acid length linkers. All linkers are repeats of a GGGGS motif, and are thus flexible. The exact length and amino acid composition of the linkers used has been added to lines 605-610.

2) Did the authors consider fusing an entire IgG to a ferritin monomer? What is the advantage of the current design? To minimize steric hindrance, could it be an advantage to only have e.g. 12 binding moieties of the surface, or is more always better?

We thank the Reviewer for his/her suggestion of making a Multabody that multimerize full IgG's instead of IgG fragments. We haven't yet explored this concept. From our perspective, the main advantages of the current design are:

- 1) Addition of IgG over Fab/Fc fragments will significantly increase the hydrodynamic radius (Rh) of the molecule beyond that found in natural antibodies (i.e. IgM), which could have undesired *in vivo* implications for example in biodistribution.**
- 2) Physical separation of the Fab and the Fc fragments allows to achieve a higher amount of Fabs compared to Fc, so that avidity of the Fab can be favored over avidity of the Fc.**
- 3) Separating Fab/Fc fragments also allows better plug-and-play features for the design.**

Furthermore, the ability to display a high number of Fabs is of critical importance particularly in the case of multi-specific molecules. Indeed, the presence of different specificities dilutes the avidity effect of each of them. It is therefore desirable to have a design where as many Fab moieties can be integrated from the start to deliver multi-specificity with avidity.

3) The authors state that the MB preparations typically have 24 Fab and 8 Fc presented – what is the experimental evidence for this distribution? Did all MB preparations of all the mentioned constructs always contain an Fc-fraction (this is not clear from the MM section)? What are typical yields of these expressions and how homogeneous are they after purification (please show examples)

We refer the Reviewer to the explanation provided on point #2 for Reviewer 1 above for a more detailed explanation on the theoretical amount of each fragment of the Multabody. As discussed, in the absence of experimental evidence to corroborate the theoretical amount, we have decided to remove such statement from Fig. 3A.

Indeed, all Multabodies described in the manuscript have an Fc component. Only the VHH72-hFerr particle from Fig. 1 does not contain Fc fragments, and is not referred as a Multabody in the text. This molecule is described in a separate section in the Materials and Methods section. To avoid any confusion with regards to the presence of the Fc domain, we have added the following sentence to line 604 of the Materials and Methods section: “All molecules referred herein as Multabodies contain single-chain Fab (scFab) and Fc (scFc) fragments”.

We have now added Fig. S3A to show the yields obtained for Multabodies in comparison to their parental IgG’s when produced in HEK 293F cells. As seen in the figure, expression yields of both antibody formats are comparable and mainly dependent on the Fab specificity. In addition, Fig. S3B shows batch-to-batch homogeneity between independent purifications for one example Multabody. For comparison of the homogeneity observed in conventional IgG’s, the parental IgG was included. Homogeneity of the purified Multabody can be observed from the monodisperse peak on the SEC chromatograms and from the similar IC₅₀ values obtained in pseudovirus neutralization for different batches. The following sentence has been added to line 187 describing this data: “Notably, MB expression yields, homogeneity and thermostability was similar to those of the parental IgG (Fig. S3)”

4) Numerous VHH high-affinity RBD binders (after immunization or from synthetic libraries) with decent neutralization efficiencies have been described. Can these ones be also formatted to the MB platform and would they have an advantage or disadvantage compared to mAB or Fabs (to be discussed in the discussion section).

Fig. 1 shows how apoferritin multimerization can enhance the neutralization properties of a VHH domain (VHH72) against SARS-CoV-2. However, as the Reviewers pointed out, this figure does not probe that VHH domains can in fact be formatted as a Multabody (i.e.

following the split design format to also incorporate Fc domains). To highlight the plug-and-play nature of the Multobody platform that also encompasses VHH domains, we made a VHH72 MB and assessed its neutralization against SARS-CoV-2 PsV and compared it to the neutralization properties of the VHH72-Fc molecule (the benchmark comparator in Fig. 1). As can be seen in the following figure, the VHH domain can be formatted as a MB and it preserves the enhanced neutralization potency observed in a 24-mer VHH72 molecule (Fig. 1E).

The use of VHH domains in the MB platform can have additional advantages as mentioned now in the Discussion on line 311: “Using the MB to enhance the potency of VHH domains could provide particular value to this class of molecules since its small size allows highly efficient multimerization.”

5) The structural data are interesting but for a “non-structural biologist” not straight forward to understand. A figure on the trimeric nature of the spike protein could help to guide the reader and illustrate the captured conformation. Furthermore, it is not stated how many Fabs are bound to the spike protein and which configuration the RBDs display (up, or down). Color-code in Fig. 5e is unclear, hard to read and understand. All structure work (X-ray and cryo-EM) has been done with Fabs and not full IgGs – this description is mixed up, please unify. Multiple structures of the Spike protein have been determined to higher resolution described here – could Fab binding lead to the preferred orientation of the complex on the EM grid requiring tilted data collection? Is this commonly seen in cryoEM with Fabs?

As suggested by the Reviewer, the trimeric unbound structure of the spike protein of SARS-CoV-2 has been added to a revised Fig. 5E for better clarity. The side and the top views are illustrated. Accordingly, the figure legend on line 464 has been re-worded to help guide the reader: “e) Composite image depicting the side and top view of the unliganded (PDB 6XM4) and the antibody-bound SARS-CoV-2 spike with available PDB or EMD entries^{3,4,9,10,13,15,17,50–55}. Inset: close up view of antibodies targeting different antigenic sites on the RBD. The mAb with the lowest reported IC₅₀ value against SARS-CoV-2 PsV was selected as a representative antibody of the bin and those antibodies with similar binding epitope have been listed in the same color below (color coding of SARS-CoV2 and RBD as in (b). Individual protomers in the unliganded spike are shown in white, pink and purple.”

To prevent Fabs from dissociating off the recombinant Spike during sample preparation for cryoEM analysis, a crosslinking agent was added. This approach allowed us to elucidate the binding site of selected Fabs. We believe that this methodology prevents a detailed analysis of the binding stoichiometry and RBD dynamics in our cryo-EM reconstructions, and have thus conservatively omitted such description.

Regarding the concept of preferred orientation of the Fab-antigen complex on the grid, the initial datasets collected displayed signs of preferred orientation and therefore additional datasets were collected under tilt. This information is included in the Materials and Methods section (line 827-831): “To overcome preferred orientation encountered with some of the samples, tilted data collection was employed⁵⁵. For the Spike-Fab 80 complex, 820 0° tilted movies and 2790 40° tilted movies were collected. For the Spike-Fab 298 complex, 4259 0° tilted movies and 3513 40° tilted movies were collected. For the Spike-Fab 324 complex, 1098 0° tilted movies and 3380 40° tilted movies were collected. For the RBD-Fab 46 complex, 4722 0° tilted movies were collected.” We speculate that molecular motion prevented visualization of the complex at high resolution. Several cryo-EM maps were published in similar resolution ranges, including reconstructions of Fab 2-43 bound to the SARS-CoV-2 Spike (Liu et al., *Nature* 584, 450–456 (2020)) and Fab S2H14 bound to the SARS-CoV-2 Spike (Piccoli et al., *Cell* 183(4):1024-1042.e21 (2020)).

We thank the Reviewer for the careful reading of the manuscript and identification of a mixed nomenclature regarding the use of Fab and mAb in the structure section. These have now been reconciled throughout the manuscript.

6) Introduce abbreviations e.g. “MB” as Multabody is not mentioned

The abbreviation MB has been added in line 51 in the following sentence: “The MULTIspecific, multi-Affinity antiBODY (Multabody or MB) platform”

7) Overview table of tested constructs with the measured affinities and neutralization efficiencies could be useful as a supplementary table.

Neutralization potency against SARS-CoV-2 PsV and replication competent SARS-CoV-2/SB2-P4-PB virus measured for the newly described IgG and MBs has now been added to Supplementary Table 1.

8) Binding Curves are called sensorgrams – I recommend adding the fitted parameters to all sensorgrams

As suggested by the Reviewer, the words “Kinetics/Binding curves” in lines 442 and 525 has been replaced by its proper name “sensograms” and the fitting model used in these experiments has been added to Materials and Methods section, line 655: “Analysis of the sensograms was performed using the Octet software, with a 1:1 fit model”.

9) What would be the next step of relevance to make use of the study?

We thank the Reviewer for his/her interest in the next steps beyond this study. We are currently finalizing selection of lead MB molecules, and are focusing on antibody specificities able to neutralize all SARS-CoV-2 variants of concerns, including B.1.1.7, B.1.351, B.1.1.28, and P1 in order to confer broad and potent neutralization with a single molecule. As described in response to Reviewer 2, a next milestone and focus of ongoing studies is the full characterization of the in vivo properties of our lead SARS-CoV-2 MBs in pre-clinical models of SARS-CoV-2 infection.

Reviewers' Comments:

Reviewer #1:

Remarks to the Author:

The authors has largely addressed the concerns in this revised manuscript. Figure 3a, however needs to be revisited as the scheme for scFc listed LC and HC as the major parts, which should not be contained in this construct.

Reviewer #2:

None

Reviewer #3:

Remarks to the Author:

The authors reworked parts of the manuscript and replied to the comments and suggestions raised by the reviewers. Additional data have been added, certain sections have been modified and all the raised concerns have been addressed properly. This revised version is ready for acceptance.